# Academic discourse on ChatGPT in social sciences: A topic modeling and sentiment analysis of research article abstracts

Yating Tao[1], Qian Shen[2]*

1 Centre for English Corpus Linguistics, Université catholique de Louvain, Louvain-la-Neuve, Belgium,
2 Department of English, North China Electric Power University, Baoding, China

* shenqian@ncepu.edu.cn

## Abstract

The rapid emergence of ChatGPT has sparked extensive academic discourse across multiple fields. This study focuses on such discourse within the social sciences by examining how scholars frame and evaluate ChatGPT through research article abstracts. Drawing on 1,227 SSCI-indexed abstracts published between 30 November 2022 and 30 November 2024, we adopt a two-step natural language processing approach. First, we apply topic modeling to identify major thematic patterns in academic discussions of ChatGPT. Then, we perform sentiment analysis to examine how scholars' evaluative attitudes are discursively constructed across these thematic areas. Topic modeling reveals six key themes: artificial intelligence (AI) and technology communication, education and learning tools, user perception and adoption, ethics and academic challenges, human-technology interaction, and computational foundations of Large Language Models (LLMs). Sentiment analysis suggests that approximately 82.97% of abstracts express positive attitudes, particularly regarding ChatGPT's research potential and pedagogical utility, while around 9.78% reflect more cautious or negative views, often focusing on issues such as academic integrity and misinformation. These sentiment patterns appear to vary across thematic areas, with user adoption and education-related topics showing greater positivity, while ethics-oriented discussions exhibit relatively more critical perspectives. By analyzing academic discourse as reflected in research article abstracts, this study contributes a discourse-level perspective on how ChatGPT is framed, endorsed, and critically examined in the social sciences. It offers a data-driven complement to existing conceptual and survey-based investigations and draws attention to both the thematic and evaluative tendencies shaping scholarly narratives around generative AI.

**Data availability statement:** The data underlying the results presented in the study are available from Figshare: https://doi.org/10.6084/m9.figshare.29625773.

**Funding:** The author(s) received no specific funding for this work.

**Competing interests:** The authors have declared that no competing interests exist.

# 1. Introduction

The emergence of generative artificial intelligence (AI) tools has reshaped how knowledge is produced, disseminated, and consumed. Among these, ChatGPT, developed by OpenAI, has drawn significant global attention since its public release on November 30, 2022. With each iteration from GPT-3.5 to GPT-5, ChatGPT has demonstrated increasingly sophisticated capabilities, ranging from code generation and automated summarization to personalized content creation. Its integration into domains such as healthcare, media, education, and academic research has been especially notable [1–4].

The integration of ChatGPT into academia has sparked extensive debate across disciplines, centering on both its potential benefits and ethical concerns. On the one hand, studies highlight its ability to enhance productivity, streamline academic workflows, and improve writing efficiency. For example, [5] and [6] emphasized its value in tasks such as language editing, data analysis, and overcoming writer's block, while [7] underscored its utility in supporting pedagogical practices and improving teaching and learning efficiency. On the other hand, this optimism is tempered by warnings about academic integrity and the risk of misuse. Some studies cautioned that generative AI tools like ChatGPT could inadvertently enable unethical academic behaviors such as plagiarism, particularly given current detection systems' limited capacity to reliably differentiate between acceptable and inappropriate uses [8,9]. Empirical findings by [10] suggested a potential correlation between increased reliance on AI and challenges to academic honesty, with convenience and cognitive ease identified as contributing factors.

These tensions have prompted calls for robust institutional policies and detection mechanisms to safeguard academic values. Against this background, scholars such as [11] and [12] advocated for proactive policy-making and institutional reforms by introducing workshops, curricular adjustments, and resources to align academic goals with the realities of AI-assisted education. [13] stressed the importance of fostering student integrity and technological self-efficacy through targeted training.

Perspectives on ChatGPT's role in academia also diverge across disciplines and stakeholder groups. [7] reported that educators generally perceive generative AI as a valuable pedagogical tool, whereas [11] and [14] highlighted more skeptical institutional stances, with some universities imposing restrictions or bans in certain cases. Additionally, disciplinary perspectives diverge in their emphasis on ethical considerations. [10] focused on students' psychological motivations for AI use, particularly convenience as a driver of unethical behavior, and [6] framed ChatGPT as a broader technological disruptor, shifting the focus from ethical concern to its broader implications for knowledge generation. In medical education [1], foregrounded ethical imperatives such as transparency and accountability in AI-assisted research and education. Taken together, these findings illustrate the complex and often contested nature of ChatGPT's integration into academic practice.

Although scholarly interest in ChatGPT is growing, many existing studies primarily adopt conceptual or qualitative approaches, such as case studies, surveys, or policy commentaries. Despite the methodological diversity of prior research, there is a

growing need for data-driven analysis to map how the academic community thematically frames ChatGPT and articulate its attitudes towards this technology. To address this gap, this study adopts an exploratory research design to analyze 1,227 SSCI-indexed abstracts from the social sciences, aiming to uncover the thematic patterns and evaluative attitudes in academic discussions of ChatGPT.

In doing so, we draw on natural language processing (NLP) techniques: Latent Dirichlet Allocation (LDA) topic modeling to identify thematic patterns, and lexicon-based sentiment analysis to capture evaluative orientations embedded in the academic discourse. In this context, sentiment analysis does not aim to detect personal emotions but rather identifies how attitude, confidence, or concern are discursively encoded in academic abstracts, treating sentiment as an indicator of evaluative positioning.

While academic abstracts are typically regarded as concise and objective summaries, research suggests that they also function as strategic rhetorical acts. From a discourse-analytic perspective, abstracts not only summarize research content, but also actively position the work by highlighting its novelty, significance, and credibility. Studies by [15–17] and [18] have shown that abstracts routinely encode authorial stance, evaluative claims, and epistemic positioning. These characteristics make them an effective genre for analyzing how emerging technologies are framed and assessed within academic discourse.

Building on this rationale, the present study applies LDA and sentiment analysis to examine how scholars engage with ChatGPT, both thematically and evaluatively. In doing so, we contribute to a broader understanding of how the academic community is responding to the challenges and opportunities presented by generative AI. Specifically, this study addresses the following research questions:

(1) What prominent thematic patterns emerge in academic abstracts on ChatGPT in social sciences?

(2) What general sentiment orientations do scholars express towards ChatGPT in these academic abstracts?

(3) How do sentiment expressions vary across the identified thematic areas in these academic abstracts?

## 2. Literature review

### 2.1. Topic modeling in text analysis

Topic modeling is a set of machine learning methods and algorithms for revealing latent thematic structures within larger corpora [19,20]. By representing each text as an integration of 'topics', clusters of co-occurring words, topic modeling can uncover thematic patterns and explore latent structures within a large number of texts. Among the most widely used methods, LDA has been instrumental in analyzing vast collections of texts to uncover thematic structures, trace the evolution of discourse, and investigate interdisciplinary connections across a broad spectrum of domains. For instance [21], demonstrated how topic modeling can measure interdisciplinarity within the National Science Foundation's portfolio, uncovering patterns of internal and external collaboration. Similarly [22], examined 25 years of professional art criticism, revealing shifts in thematic priorities towards societal and political concerns. In education [23], highlighted emerging trends in educational technologies, such as e-learning and game-based learning. These studies demonstrate the feasibility of topic modeling as a data-driven approach for exploring, categorizing, and interpreting latent themes in unstructured text.

Even though topic modeling is powerful on its own, it is often combined with other methods to provide a more systematic, rigorous, and insightful analysis of the texts. One notable example is its combination with critical discourse analysis (CDA), allowing researchers to interpret each topic in a more specific way. In a recent study [24], employed topic modeling and critical discourse analysis to examine news reports on Chinese migrant workers. Their analysis identified key topics related to work, rights, and social challenges, revealing various representations of migrant workers in the media. [25] argued that combining topic modeling with qualitative approaches broadens the scope of discourse analysis by identifying

patterns of hegemony in language use. Likewise [26], employed topic modeling to connect discursive threads between Islamophobia and anti-feminism in online forums. In addition to studies on social and political issues [27], examined climate change discourse, shedding light on the interplay between energy and food security narratives. These studies illustrate the versatility of topic modeling in uncovering thematic structures in a wide range of textual data.

### 2.2. Integration of topic modeling and sentiment analysis

Of particular relevance to this study is the integration of topic modeling and sentiment analysis. Topic modeling identifies the latent topics across unstructured texts, which can reveal key debates or concerns around a topic, whereas sentiment analysis evaluates people's general attitudes and emotions within those discussions [28]. This combined approach has garnered great attention across different fields. [29], for example, applied this approach to predict stock market price movements based on social media sentiments. [30] used this approach to identify key aspects of consumer reviews on airline services. Similarly, other researchers examined public sentiments on global warming and environmental issues [31–33].

In addition, diverse approaches to sentiment analysis have been proposed to explore the evaluative dimensions of language use in different contexts [34,35]. These models have demonstrated high accuracy and have been employed to analyze large-scale and real-time data (e.g., tweets) to assess public opinions on dynamic events such as global warming debates [32] and Uber services [36]. More recent studies have further enhanced aspect-based sentiment analysis by reframing it as a masked language modeling task for improved affective token prediction [37,38] and by integrating descriptive lexical knowledge from resources like the Oxford Dictionary to enrich semantic representation [39]. These studies demonstrate the methodological versatility of sentiment analysis, along with its increasing integration with topic modeling, in detecting sentiment patterns from large-scale, real-time textual data across diverse discourse domains.

Additionally, topic modeling and sentiment analysis have proven effective in capturing themes and evaluative patterns in academic texts. For example, topic modeling has been used to classify thematic structures in scholarly abstracts [40] and to trace disciplinary trends over time [41]. Sentiment analysis has similarly been applied to detect evaluative signals in randomized trial abstracts [42], full-length research articles [43], peer reviews [44,45], and citation contexts [46]. These studies demonstrate that even in highly structured and ostensibly neutral texts, such as academic abstracts, evaluative attitudes can be empirically detected. This methodological precedent supports the application of topic modeling and sentiment analysis to investigate academic discourse in this study.

### 2.3. Public attitudes towards chatbots

#### 2.3.1. Public attitudes in social media.
A growing number of studies have integrated topic modeling and sentiment analysis to analyze public attitudes towards ChatGPT and similar chatbots across various social media including Twitter [2,47–49], Weibo [50,51], and Reddit [52]. These studies reveal a complex interplay between enthusiasm for technological advancements and concerns about privacy, ethical issues, and societal implications. Generally, positive sentiments dominate discussions on Twitter, where many users express excitement about chatbots' applications in education and automation [48,49]. Specifically, chatbots are seen as being valuable for their potential as personalized learning tools and improving workflow efficiency. However, negative sentiments also emerge, particularly in studies addressing ethical concerns, such as academic integrity [2] and ethical issues [52]. [2] observed significant skepticism surrounding ethical and practical concerns. Likewise [52], identified that negative comments on Reddit reflect widespread apprehension about ChatGPT's limitations, inaccuracies, and ethical implications. These skeptical views are further reflected in broader concerns about data security and societal risks. [53] identified data privacy as a central topic in discussions about ChatGPT, and [47] tracked the evolution of these concerns over time, demonstrating how users' attitudes shift in response to policy changes.

#### 2.3.2. Professional perspectives on ChatGPT.
Beyond the sentiment trends in social media, studies also reveal professional variations in attitudes towards ChatGPT. [3] noted higher enthusiasm among tech professionals, compared

to educators and policymakers, who express concerns about the potential impacts of ChatGPT on academic integrity and ethical use in education. Education is a key domain in discussions about the role of ChatGPT, reflecting its transformative potential and ethical challenges. Studies such as [48] and [49] emphasized the benefits of ChatGPT in personalized learning, study assistance, and creative assessments, framing it as a valuable tool for students and educators. However, educators are also concerned about academic integrity and misuse, and the impact on learning outcomes and skill development [2,49]. In addition, recent studies have explored ChatGPT's role in healthcare. [54] reported mixed sentiments in an exploratory survey across education, healthcare, and research, with 40% of participants reporting ChatGPT usage and expressing both optimism and ethical concerns. [55] applied ChatGPT and aspect-based sentiment analysis to over 500,000 patient reviews, illustrating its potential to extract healthcare needs from patient perspectives. These divergent professional perspectives from tech enthusiasm to educational caution to healthcare ambivalence demonstrate the critical need for sector-specific implementation strategies that address distinct professional requirements and ethical considerations.

Despite the increasing attention to public and occupational perceptions of ChatGPT across different regions and professional domains, it remains unclear how these developments have influenced attitudes within the academic community. While several studies have investigated academic perspectives on ChatGPT in social science through qualitative approaches [56–59], few have empirically examined how this technology is thematically framed and evaluated in academic discourse. To address this gap, the present study applies topic modeling and sentiment analysis to analyze how ChatGPT is framed and evaluated in academic abstracts from the social sciences.

## 3. Data and methodology

### 3.1. Data collection and cleaning

A total of 1,227 abstracts of research articles were collected from SSCI-indexed journals available in the Web of Science (WoS) database. The dataset spans a broad range of disciplines within the social sciences, including but not limited to psychology, education, linguistics, communication studies, and sociology. This study focuses on the social sciences not only because they provide a critical lens for examining how ChatGPT reshapes human behavior and social systems, but also because their interdisciplinary nature generates a diverse and dynamic body of academic discourse for analyzing emerging perspectives and sentiments towards generative AI.

While our dataset covers a wide spectrum of social science disciplines, we did not further disaggregate it by subfields. This decision aligns with our overarching research goal to provide a comprehensive, cross-disciplinary overview of the ways in which social science scholarship engages with ChatGPT. Given the exploratory nature of the study, our study is not to conduct comparative analyses across individual subfields, but rather to identify general thematic and evaluative patterns emerging from the broader academic discourse. We acknowledge that disciplinary variation may exist and consider this a valuable direction for future research.

We used "ChatGPT" OR "generative AI" as keywords to search the SSCI-indexed WoS database. The search was restricted to publications between 30 November 2022 and 30 November 2024. This initial search yielded 1,421 records, which were exported in Excel format and subjected to manual screening. To ensure data quality and conceptual relevance, records underwent a two-step screening process. First, 66 entries lacking abstracts or containing placeholder text (e.g., "No abstract available") were excluded. Second, 128 non-academic entries (e.g., editorials, book reviews), non-English abstracts, or entries lacking meaningful content were removed. After filtering, 1,227 abstracts remained and were converted into CSV format and imported into R [60] for subsequent analysis.

Systematic data cleaning and preprocessing were then conducted using R packages. All abstracts were first converted into lowercase to ensure uniformity, and all punctuation, numbers, and special characters were removed. Common English stopwords (e.g., *the*, *and, is*) were eliminated as they do not contribute meaningful information for the analysis.

Lemmatization was applied to reduce words to their base forms (e.g., convert *using* to *use*), ensuring uniformity and enhancing analytic quality. Following the cleaning process, the abstracts were tokenized into individual words, breaking the abstracts into smaller units and facilitating detailed textual analysis.

After preprocessing, we employed a two-step natural language processing workflow. First, topic modeling uncovered prominent themes in the corpus. Second, sentiment analysis, employing the NRC Emotion Lexicon, examined how evaluative attitudes towards ChatGPT are articulated in academic discourse, treating sentiment as a discursively encoded scholarly positioning.

### 3.2. LDA

To identify key themes hidden within the datasets, this study employs LDA, an unsupervised probabilistic topic modeling technique. LDA enables dimensional reduction of complex textual data into interpretable semantic spaces through a sophisticated three-layer Bayesian hierarchical model with an estimation of a sparse Dirichlet prior. LDA assumes each document as a multinomial distribution over topics and each topic as a multinomial distribution over words [61].

The generative process of LDA in Fig 1 illustrates how topics are assigned to words within documents through a three-layer Bayesian hierarchical model. At the document level, each document d follows a multinomial distribution $\theta_d$ over K topics, where the distribution is governed by a Dirichlet prior, denoted by the parameter α. The Dirichlet distribution is a probability distribution over multinomial distributions, ensuring that the topic probabilities sum up to one. The α parameter plays an important role in topic allocation. A higher α value promotes topic diversity while a lower value encourages sparsity. This distribution effectively quantifies the thematic composition of individual documents within the corpus. At the word level, for each word i in document d, LDA assigns a topic $z_{d,i}$ based on the document's topic distribution ($\theta_d$). For each topic $z_{d,i}$, LDA draws a distribution over words ($\varphi_z$), governed by the parameter β, which controls the distribution of words within a topic. A higher β value leads to more topic overlap while a lower β value makes topics more distinct. The topic assignment and word generation processes are refined through several iterations until the model reaches a stable state. Through this iterative optimization, LDA ultimately reveals the hidden thematic patterns of the dataset. In our implementation of the LDA model, we adopt commonly recommended hyperparameter values to ensure stable and interpretable results: α is set to 50/K, where K is the number of topics, and β is set to 0.1. These settings are in line with standard practices suggested by [61] and are known to balance topic sparsity and generalization effectively across diverse corpora.

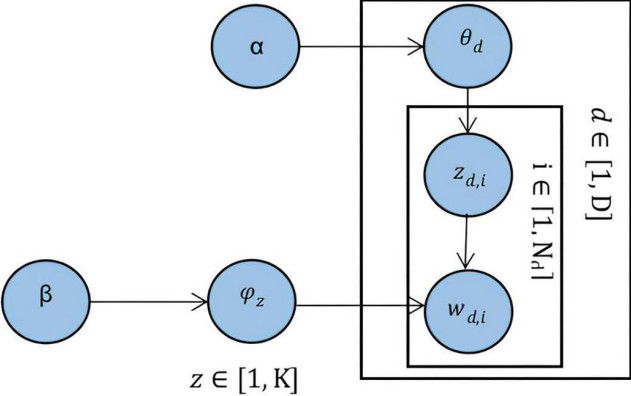

**Fig 1. Generative process of LDA (adapted from [61]).**

In our study, we conducted LDA in R using the *topicmodels package* [62]. Before performing LDA topic modeling, it is necessary to decide on the optimal number of topics to ensure a balance between model interpretability and topic coherence. This balance is essential because too many topics can create overlapping, redundant categories that confuse interpretation, while too few topics may oversimplify by combining distinct concepts into broad categories. To do so, we employed four metrics from the R package *ldatuning* [63], namely Arun 2010 [64], CaoJuan 2009 [65], Deveaud 2014 [66], and Griffiths 2004 [67]. The Arun 2010 metric measures topic sparsity by evaluating the distribution of topics across documents. A lower value suggests an appropriate number of topics that aligns well with the data. The CaoJuan 2009 metric assesses topic similarity, aiming to minimize overlap and ensure distinctiveness among topics. The lowest value indicates the optimal number of topics. The Deveaud 2014 metric evaluates semantic coherence within topics, reflecting the quality and meaningfulness of topic-word associations. A higher value indicates better coherence, making maximization the goal. The Griffiths 2004 metric measures the model's likelihood of generating the observed data. Higher likelihood values imply better model fit, also requiring maximization. By considering these four metrics collectively, we were able to make a robust and informed decision about the optimal number of topics, striking a balance between distinctiveness and coherence. Once the model was optimized, the top 10 keywords with the highest probability within each topic were extracted and analyzed to assign meaningful labels that reflect the key topics of the academic discourse on ChatGPT. This systematic approach ensured that the identified topics were both coherent and meaningful, offering valuable insights into how scholars perceive ChatGPT in social sciences.

### 3.3. Sentiment analysis

Sentiment analysis aims to identify and extract subjective information and emotions from texts by automatically classifying texts as positive, negative, or neutral. Sentiment analysis can reveal the attitudes, opinions, and feelings expressed in various forms of communication, such as customer reviews, social media posts, and news articles. Within the field of sentiment analysis, various approaches or techniques are used for different purposes including machine learning techniques [68–70], deep learning architectures [71–73], and lexicon-based approaches [74–76].

A lexicon-based approach, specifically the NRC Word-Emotion Association Lexicon (NRC EmoLex) [77], is adopted in this study to examine scholars' evaluations towards ChatGPT in academic settings. The NRC EmoLex is a comprehensive resource categorizing words based on eight emotions (anger, fear, anticipation, trust, surprise, sadness, joy, and disgust) and two sentiment polarities (positive and negative). Instead of relying solely on experts, this lexicon was created using crowdsourced annotations, which means that the creators gathered data from a large number of people online who labeled the emotional associations of words, thus making it a robust tool for sentiment analysis. Although the NRC lexicon was originally developed for general-purpose and social media texts, it has been adopted by [78] to score sentence-level positivity/negativity in life science abstracts. While we acknowledge that its application to academic abstracts may not fully capture the subtlety of sentiment expression in academic discourse, this limitation is less significant in our case. Our goal is not to detect affective cues at the sentence level, but rather to uncover broader evaluative tendencies across academic abstracts. From this perspective, the NRC lexicon remains a useful and interpretable tool for sentiment analysis across documents. The R package syuzhet [79] was used to generate sentiment scores for each abstract by counting the frequency of words associated with different emotions and sentiments in this study.

## 4. Results and discussion

### 4.1. A general overview of the data

To get a general overview of the datasets before the main analysis, we first extracted the top ten most frequent words related to ChatGPT in the collected abstracts, as shown in Fig 2. This distribution of key terms demonstrates the predominant research focus and theoretical orientation surrounding ChatGPT in the social sciences.

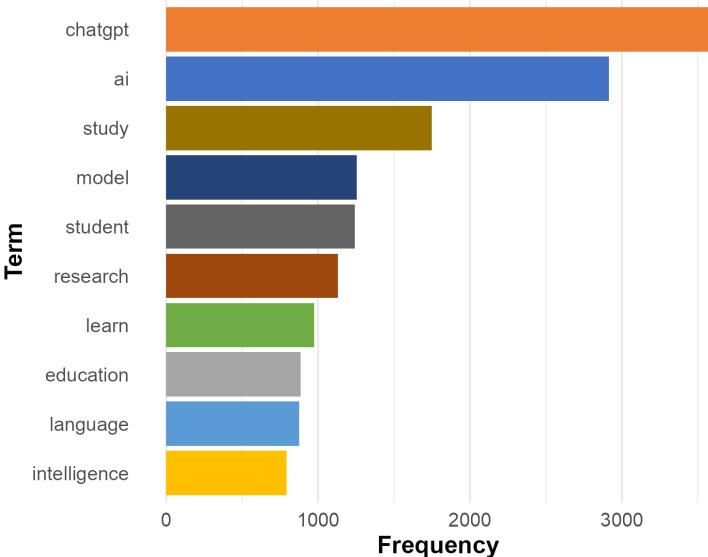

**Fig 2. Ten most frequent terms.**

The prominence of "*ChatGPT*" and "*ai*" confirms their central roles as the primary subjects of inquiry, which is crucial for understanding the scope and potential impact of ChatGPT across various domains. The second cluster emerging from the top ten most frequent words consists of "*study*", "*research*", and "*education*", which suggest that educational contexts may dominate the current scholarly discourse in social sciences. This finding aligns with the broader academic interest in understanding the pedagogical implications and educational applications of ChatGPT. The presence of technical and methodological terms such as "*model*", "*language*", and "*intelligence*" indicate a robust theoretical and technological discussion surrounding ChatGPT. This terminology cluster reflects the interdisciplinary nature of ChatGPT research, encompassing both pedagogical applications and technical implementations.

While the frequent appearance of terms such as "*study*", "*research*", and "*education*" in Fig 2 may partly stem from the generic conventions of abstract writing, it also suggests a meaningful research focus within the social sciences, especially in fields like education, where scholarly engagement with ChatGPT appears to be particularly active. The appearance of "*student*" and "*learn*" among the top 10 most frequent words further implies that researchers are particularly concerned with learning processes and the impact of ChatGPT on students' learning outcomes. This interpretation is supported by the co-occurrence network shown in Fig 3, generated using the LancsBox X [80]. The central node, "*student*", serves as the focal point, with node size reflecting co-occurrence frequency and connecting lines illustrating relationships within the corpus. The strong association of "*student*" with "*learning*", indicated by the large node size and prominent connection, highlights the centrality of learning-related discussions in the academic discourse around ChatGPT. This may point to a growing interest in understanding how generative AI tools like ChatGPT influence educational experiences and learning outcomes.

## 4.2. LDA topic modeling analysis

**4.2.1. Optimal number of topics.** While the top 10 most frequent words provide a general idea about scholars' attention in social science, they cannot offer a deeper and nuanced understanding of the hidden themes depicted in the abstracts. In this aspect, LDA can be used to uncover the prominent topics. The optimal number of topics was determined based on four established metrics from the R package ldatuning, i.e., Arun 2010, CaoJuan 2009, Deveaud 2014, and

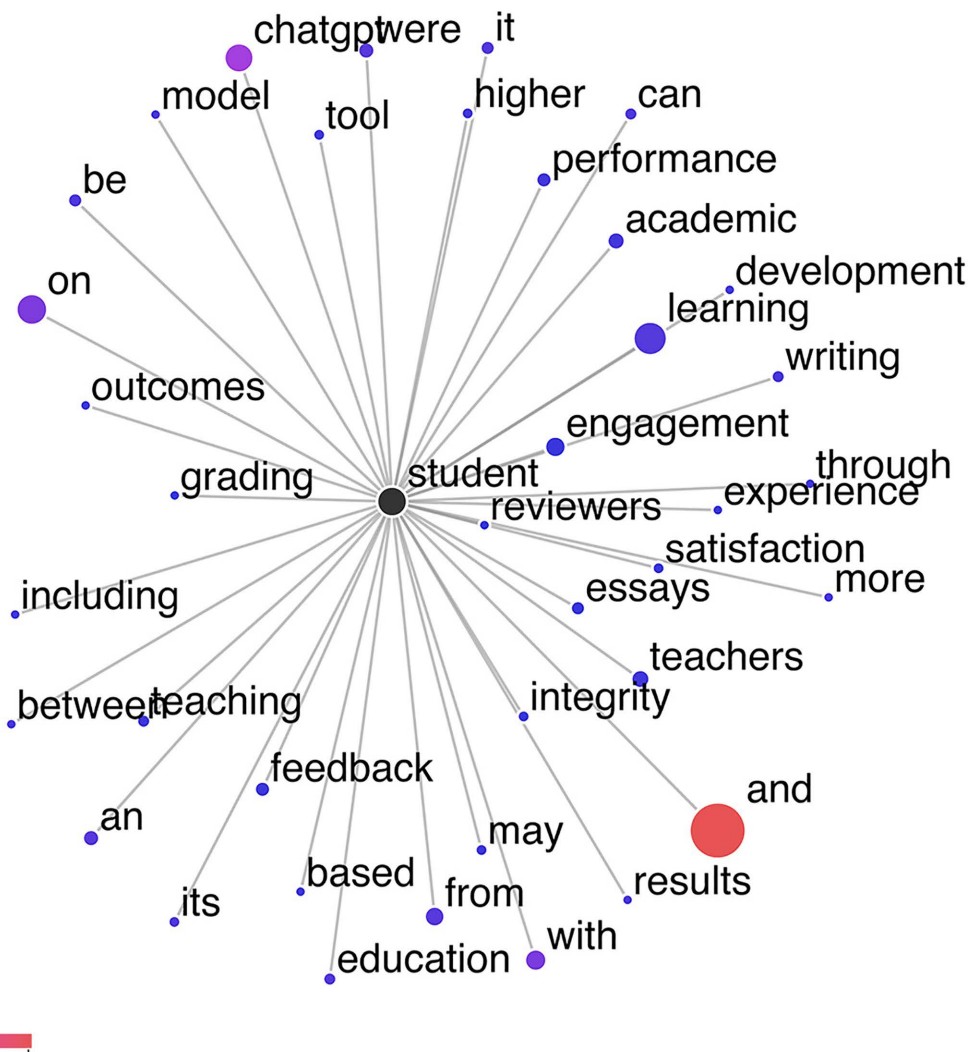

**Fig 3. Collocation graph of "student".**

Griffiths 2004. Based on the comprehensive evaluation of four established metrics shown in Fig 4, the selection of six topics emerges as the optimal choice through systematic convergence analysis. The decision-making process involved identifying the point where minimization metrics (Arun 2010 and CaoJuan 2009) achieve sufficiently low values while maximization metrics (Deveaud 2014 and Griffiths 2004) reach satisfactory performance levels.

At six topics, we observe a critical convergence pattern across all four metrics. The Arun 2010 metric decreases dramatically from 1.0 at two topics to approximately 0.3 at six topics, representing a 70% reduction that indicates effective topic separation. Similarly, CaoJuan 2009 follows a steep downward trajectory, dropping from 0.85 to 0.2, demonstrating minimal topic overlap and enhanced distinctiveness. Both minimization metrics show substantial improvements from two to six topics, with diminishing returns beyond this point. The maximization metrics in the lower panel of Fig 4 shows Deveaud 2014 reaches its absolute peak of 1.0 at six topics, indicating optimal semantic coherence within topic structures. This peak is particularly decisive, as the metric declines sharply to 0.6 at seven topics, signaling that additional topic

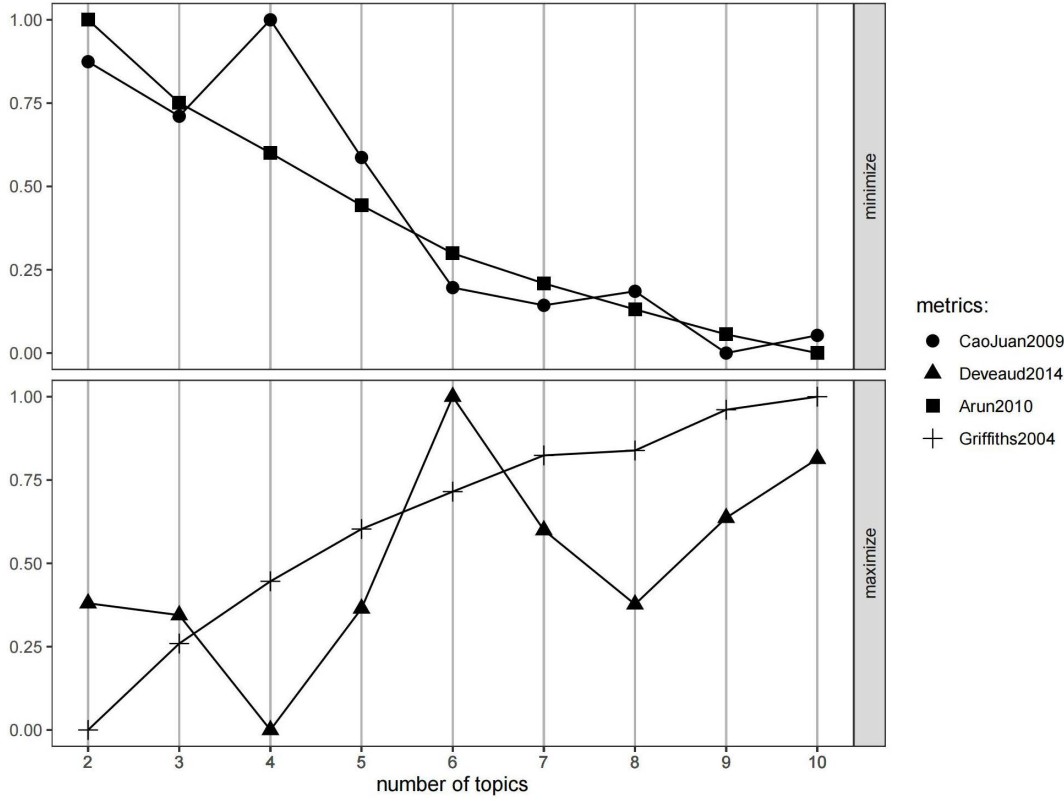

**Fig 4. Metrics for determining the optimal number of topics.**

subdivision compromises semantic meaningfulness. While Griffiths 2004 continues to increase gradually beyond six topics, reaching approximately 0.85 at ten topics, this improvement comes at the cost of semantic coherence as evidenced by Deveaud 2014's decline.

Our selection of six-topic solution can be further validated by the coherence trajectory shown in Fig 5. The coherence score evaluates how well words within each topic relate to one another semantically, which means higher scores indicate stronger intra-topic word relationships and better topic interpretability. This process involves identifying the peak coherence value, which represents the point where topics achieve maximum semantic coherence without over-fragmentation. In Fig 5, the coherence score reaches its highest value of 0.382 at six topics, suggesting that this configuration produces the most semantically coherent and interpretable topic structure. Beyond this peak, additional topic subdivision would likely result in diminished coherence as meaningful themes become artificially fragmented into less coherent sub-categories.

**4.2.2. Main topics.** Fig 6 shows the most representative terms for each topic identified through LDA. Each panel corresponds to a specific topic, and the terms are listed in order by their beta values, which quantify the importance of each term within its topic and provide a statistical basis for interpreting the topics and their key terms. In what follows, we provide a description of each topic with examples.

Topic 1 is denoted as **AI and technology communication**. It emphasizes terms such as "*ai*", "*generative*", "*intelligence*", "*artificial*", "*human*", and "*technology*", demonstrating that the academic discourse centers on artificial intelligence, its generative capabilities, and its technological dimensions. The prominence of words like "*system*", "*social*", "*public*", and "*communication*" reveals a broader focus on how AI tools like ChatGPT interact with societal systems and communication networks. Additionally, the term "*literacy*" suggests an emerging interest in how public understanding and technological

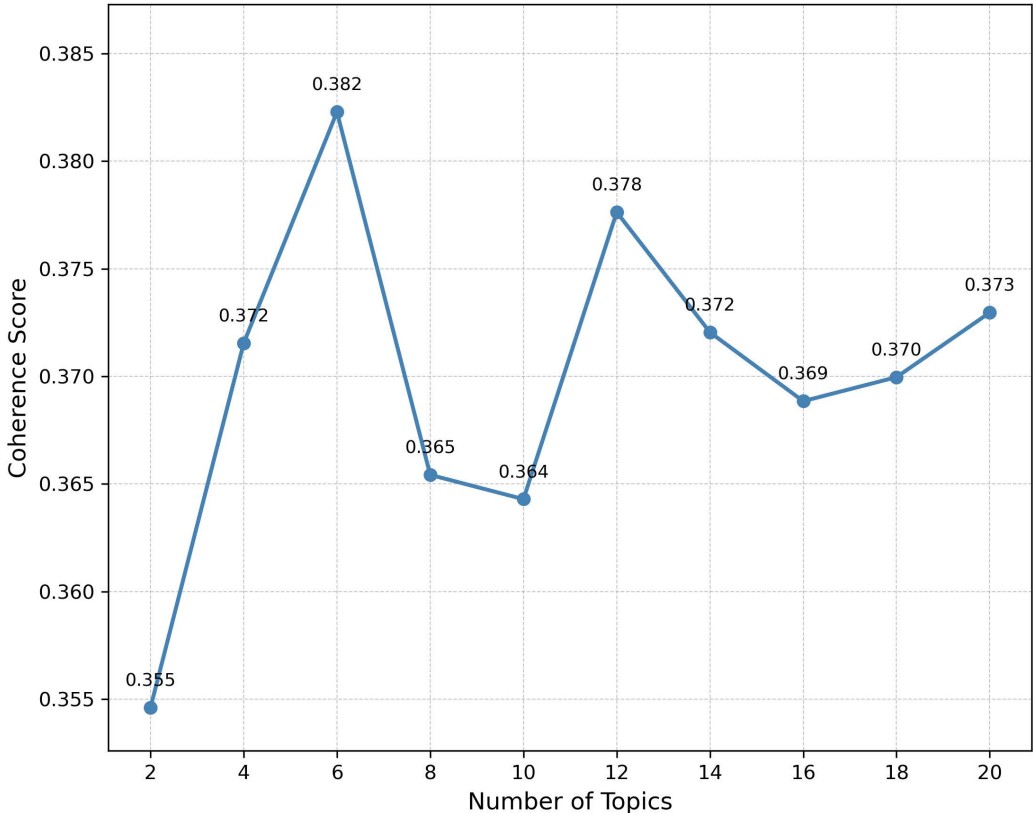

**Fig 5. Coherence score across different numbers of topics.**

literacy shape AI adoption. The following examples from the collected abstracts examine the interaction between human and artificial intelligence technology.

1. *With the vigorous development of ChatGPT and its application in the field of education, a new era of the collaborative development of **human** and **artificial intelligence** and the symbiosis of education has come.*

2. *ChatGPT enhances digital accessibility, **performance in communication**, and supports the creation of digital content, which can be a powerful **assistive technology** for the education industry.*

3. *...ChatGPT demonstrates promise in automating the analysis of **public feedback**, offering substantial time and cost savings...*

4. *...underscores the growing importance of integrating **AI literacy** into educational curricula to optimize the reception and utilization of...*

Topic 2, named **education and learning tools**, highlights the role of ChatGPT as a transformative tool in educational settings. Terms such as "*student*", "*learn*", "*study*", "*write*", "*education*", and "*teacher*" emphasize applications of ChatGPT in teaching and learning processes. The inclusion of terms like "*design*", "*enhance*", "*feedback*", and "*tool*" indicates that researchers are particularly interested in how ChatGPT can improve instructional design, facilitate personalized feedback, and enhance educational outcomes. Furthermore, the occurrence of "*assessment*", "*skill*", and "*university*" points

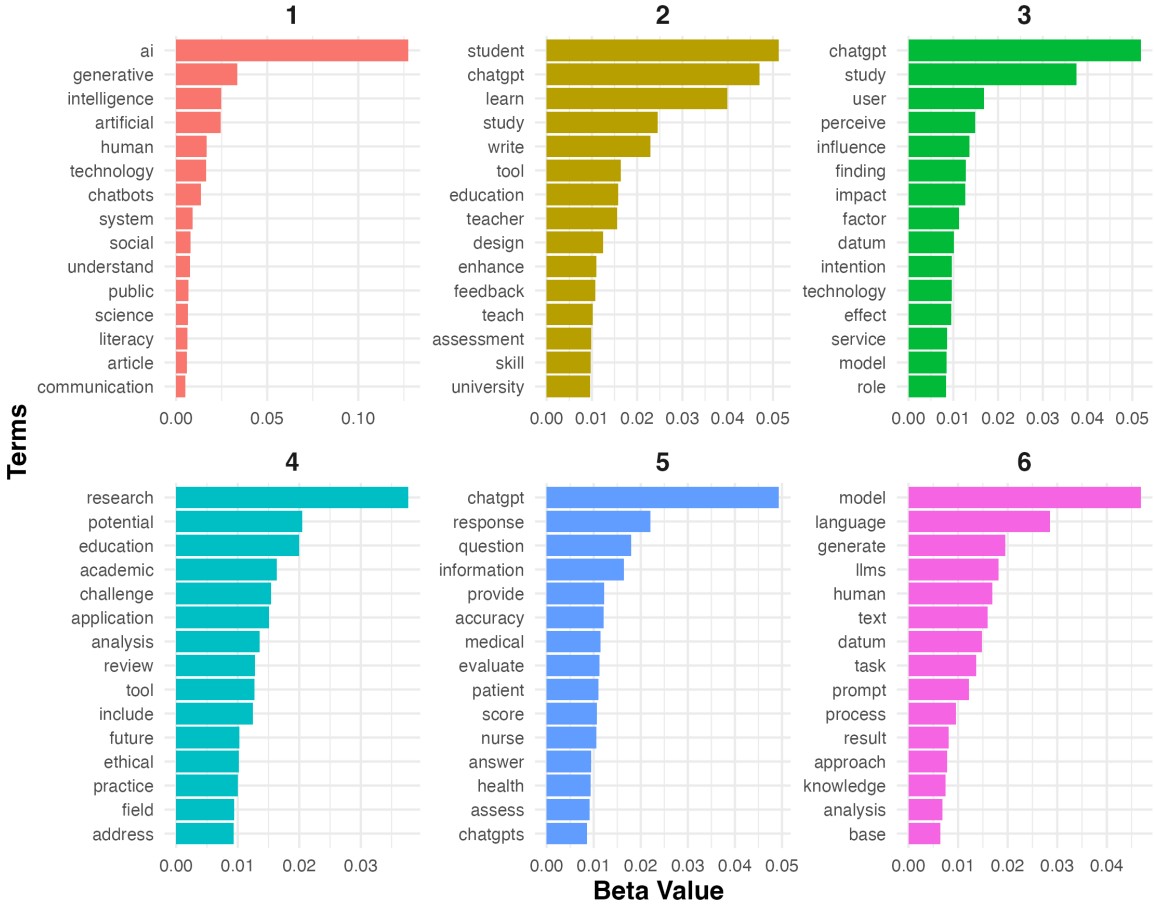

**Fig 6. Top terms for each topic.**

to ChatGPT's potential in higher education for skills development and assessment processes. The following examples demonstrate how ChatGPT is employed in educational contexts, particularly highlighting the interplay between teacher, students, and learning process.

5. *ChatGPT, a language model developed by OpenAI, is one such tool that can deliver individualized recommendations to students, increase collaboration and communication, and improve **student learning** results.*

6. *...ways that **teachers** can **teach students** to assess ChatGPT's output with a critical eye and use the tool to overcome writers' block and other obstacles to **writing**...*

7. *...these highlight the potential of ChatGPT to revolutionize pedagogical practices, **enhance course materials**, and serve as a socially immersive and interactive...*

8. *...ChatGPT can proficiently accomplish these tasks, offering potential benefits to **educational design** and **assessment** processes.*

Topic 3, defined as **user perception and adoption**, examines how individuals perceive, interact with, and integrate AI technologies like ChatGPT. Terms such as "*study*", "*user*", "*perceive*", "*influence*", and "*intention*" reveal a focus on user behavior, trust, and willingness to adopt generative AI tools across various domains. This involves behavioral research

into attitudes, trust and willingness to integrate generative AI tools into various domains. Additionally, words like "*model*", "*technology*", "*impact*", "*datum*", and "*effect*" suggest a data-driven approach to evaluating AI performance and its impact. The presence of terms such as "*role*" and "*service*" further indicates interest in ChatGPT's applicability in service-based industries and its potential to enhance user experiences. The following examples demonstrate how users perceive and adopt ChatGPT in various contexts, indicating the interplay between user perceptions, social influences, and behavioral outcomes.

9. *Advancements in AI, such as generative AI including ChatGPT and Midjourney, enhance learners'* **perceived** *interactivity, thereby facilitating learning through AI-enabled interaction.*

10. **Social influence** *and perceived value emerge as key determinants of* **user** *cognitive appraisals of ChatGPT's expertise, trustworthiness, and emotional connections through parasocial interaction.*

11. *... the use of ChatGPT has a significant indirect* **impact on students**' *research skills through the mediating variables. This suggests that high autonomous motivation and self-directed learning are crucial for students to fully benefit from ChatGPT in developing research skills.*

12. *Recognizing the potential of ChatGPT in enhancing* **customer service** *and operational efficiency is crucial for tourism enterprises.*

Topic 4, **ethics and academic challenges**, addresses critical ethical and academic considerations surrounding ChatGPT, as reflected in terms such as "*research*", "*potential*", "*education*", "*challenge*", and "*application*". This topic primarily focuses on the ethical implications of AI, particularly in academic and research contexts. Key terms such as "*ethical*", "*future*", "*practice*", "*review*", and "*field*" suggest that scholars are concerned with AI's role in reshaping research methodologies, educational practices, and disciplinary standards. The inclusion of "*analysis*" and "*address*" may further indicate broader debates about the future of AI ethics and the moral responsibilities of integrating ChatGPT into critical areas like education and research. The following examples illustrate how ethics and academic challenges manifest in various academic contexts, particularly highlighting the tension between technological potential and ethical considerations.

13. *...human touch in services, challenges to privacy and security, and the* **potential negative outcomes** *affecting service consumers and employees in terms of inequality, biases, and misuse of ChatGPT.*

14. *...Generative AI tools* **challenge academic integrity**, *pose a challenge to validating information accuracy, and require strategies to ensure the credibility of AI-generated information.*

15. *...call on the profession's governing organizations to develop a comprehensive* **ethical framework** *for the use of LLMs such as ChatGPT in* **social work research**.

16. *The potential applications of LLMs in health care education, research, and practice could be promising if the associated valid* **concerns** *are proactively examined and addressed.*

Topic 5, named **human-technology interaction**, is characterized by terms such as "*response*", "*question*", "*information*", "*accuracy*", "*evaluate*", and "*medical*", suggesting a strong focus on ChatGPT's human-AI interactions. Terms like "*patient*", "*nurse*", "*score*", and "*health*" underscore the growing interest in AI's applications in healthcare. Researchers may explore ChatGPT's accuracy and limitations in certain contexts, its ability to provide reliable information, and the ethical implications of deploying AI tools in sensitive, human-centered fields like medicine. The presence of "*assess*" and "*evaluate*" indicates a focus on the evaluation of ChatGPT's performance, which might raise questions about its role in improving accuracy and efficiency in professional tasks. The following excerpts

exemplify human-technology interaction, particularly focusing on the evaluation of ChatGPT's capabilities in medical settings.

17. *...that a majority of participants were familiar with ChatGPT and believed in its ability to understand and **respond to user queries**. They also had confidence in the **accuracy of information** provided by ChatGPT, indicating a moderate level of trust.*

18. *Integrating artificial intelligence (AI) into **medical education** has the **potential** to revolutionize it. Large language models, such as ChatGPT, can be used as virtual teaching aids to provide students with individualized and immediate medical knowledge..*

19. *ChatGPT has become a crucial tool for **assessing mental health** and emotional awareness and helping to predict outcomes.*

20. *...**evaluates** the effectiveness, reliability and safeness of ChatGPT in assisting patients with mental health problems, and to **assess** its potential as a collaborative tool for mental **health** professionals...*

Topic 6, **computational processes and foundations of LLMs**, focuses on the underlying mechanisms of language models. Key terms like "*language*", "*model*", "*generate*", "*LLMs*", "*human*", and "*text*" suggest an emphasis on language generation and NLP technology. In addition, terms such as "*task*", "*prompt*", "*analysis*", "*process*", "*result*", and "*knowledge*" point to the evaluation of ChatGPT's ability to handle prompts, generate meaningful outputs, and support knowledge-based systems. Words like "*approach*" and "*datum*" further highlight investigations into model architectures, fine-tuning strategies, and the role of data in optimizing LLM performance. The following examples illustrate how these key terms manifest in concrete research contexts, showcasing the intersection between theoretical frameworks and practical implementations.

21. *...superior performance in binary classification tasks, such as distinguishing **human-generated text** from a specific **LLM**, compared with the more complex...*

22. *...including personalized interactive learning, language support, writing and research assistance, **task** and concept clarification.*

23. *Consequently, by acquiring proficiency in **prompt** engineering, students can maximize the positive impact of ChatGPT...*

24. *...patterns in ChatGPT's output to its processing of vast **training data** and underlying statistical algorithms.*

**4.2.3. Network analysis of topics.** The topic network in Fig 7 reveals patterns of connections between specific topics. Most notably, topic 2 (education and learning tools) serves as a central hub with robust connections to both topic 3 (user perception and adoption) and topic 4 (ethics and academic challenges). This dual connectivity underscores the complexity of educational applications, where successful implementation depends critically on user acceptance while simultaneously navigating significant ethical considerations such as academic integrity and assessment fairness. Additionally, the strong link between topic 3 and topic 6 (computational processes and LLMs) demonstrates how technical performance directly influences user attitudes and adoption decisions, establishing a clear pathway from technological capabilities to practical acceptance.

The connection pattern may suggest a three-tier research architecture within ChatGPT studies: a technical foundation layer (topic 6) that provides underlying capabilities, a user acceptance layer (topic 3) that mediates between technology and application, and an application practice layer (topic 2 and topic 4) where technological potential is realized alongside corresponding challenges [81]. The positioning of user perception research as a pivotal connector between

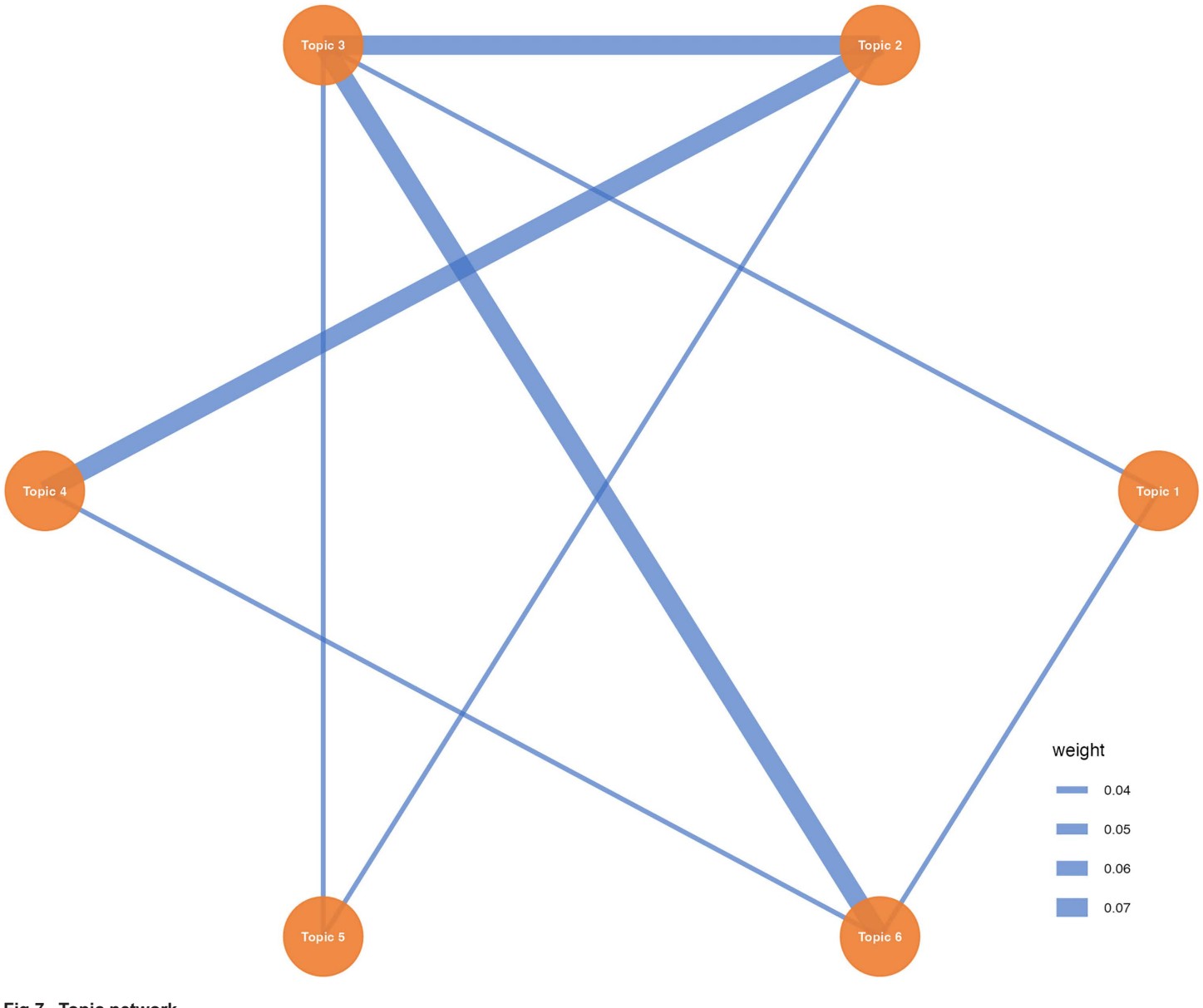

**Fig 7. Topic network.**

technical development and practical implementation reflect the field's evolution from early technical validation towards a more mature focus on real-world effectiveness and societal impact. This interconnected structure indicates that ChatGPT research is more likely to transcend disciplinary boundaries to form an integrated ecosystem where technological, behavioral, and ethical considerations are fundamentally intertwined [82].

**4.2.4. Discussion of LDA-derived topics.** To further contextualize our topic modeling findings, this section discusses each identified theme in relation to prior studies. This comparison aims to highlight points of convergence and divergence between academic and public concerns regarding generative AI. In our study, topic 1 and topic 5 underscore the academic community's significant focus on the social and technical dimensions of ChatGPT. This aligns with [52]'s identification of "socioeconomic influence" and "philosophical discussions" as key themes in public discourse on ChatGPT. The

convergence of these themes across academic and public spheres suggests a broader societal engagement with AI's impact on human-technology relationships. Notably, the persistent emphasis on human-centered perspectives, both in academic contexts and public discussions, reflects a growing recognition of the need to understand AI's social implications beyond purely technical considerations.

The prominence of education-related themes in our analysis (topic 2: education and learning tools; topic 4: ethics and academic challenges) closely corresponds with [48]'s findings on educational discourse in social media. While our analysis revealed a focus on pedagogical applications and institutional integration [48]'s, identification of financial concerns as a major cluster suggests a potential misalignment between academic research priorities and public anxieties regarding educational accessibility in the AI area. This discrepancy underscores the need for further investigation into the socioeconomic implications of AI adoption in education, particularly regarding its influence on equality and access.

The ethical dimensions (topic 4) identified in our analysis intersect significantly with the security and privacy concerns highlighted by both [47] and [53]. While academic literature tends to frame ethical considerations within broader institutional and philosophical contexts, public discourse focuses more specifically on cybersecurity threats and data privacy [53]. This difference suggests a need for a greater integration between theoretical ethical frameworks in academia and practical security concerns in public discourse.

Although [52] identified "general user impression" as a primary topic in public discourse, our findings of user perception and adoption (topic 3) extend this understanding by revealing more diverse dimensions of user interaction, particularly concerning trust formation and domain-specific patterns. In addition, our identification of computational processes and foundations (topic 6) complements [52]'s finding on "technical implementation" as a major topic in public discussion. However, while public discourse tends to center on practical applications, our findings demonstrate that academic discussions adopt a more sophisticated theoretical orientation, focusing on the fundamental architectural principles and computational paradigms in language model development. This distinction highlights the complementary nature of public and academic discourse, where practical concerns and theoretical foundations together contribute to a more comprehensive understanding of AI's evolution.

## 4.3. Sentiment analysis

Having discussed six major topics about ChatGPT in the social sciences, our sentiment analysis explores the evaluative orientation of academic discourse towards the technology. Specifically, it examines the distribution of eight emotions and the sentiment polarity derived from 1227 abstracts.

**4.3.1. Distribution of emotions and sentiment polarity in academic discourse.** Fig 8 presents the distribution of eight basic emotions: trust, anticipation, joy, fear, surprise, sadness, anger, and disgust. The emotion trust is the most dominant, with a frequency of approximately 8,400, which may reflect a generally positive and confident tone in the academic discourse on ChatGPT. This finding suggests that researchers often view ChatGPT's applications and implications with a sense of reliability and credibility. Anticipation follows as the second most frequent emotion, with roughly 3,800 instances, pointing to a forward-looking perspective where researchers appear to express interests and expectations for ChatGPT's potential, particularly in areas like natural language processing and code generation. Joy ranks third, with about 3,000 occurrences, which could be interpreted as a sign of optimism and enthusiasm about ChatGPT's contributions to various domains, such as education and technology.

On the other hand, emotions such as fear, surprise, sadness, anger, and disgust are considerably less frequent. "Fear" emerges as the most prominent among these, with approximately 2,200 instances, potentially indicating concerns about potential risks such as plagiarism, bias in outputs, or the spread of misinformation. The moderate presence of surprise and sadness (approximately 1,500 instances) may reflect occasional unexpected insights or discoveries related to

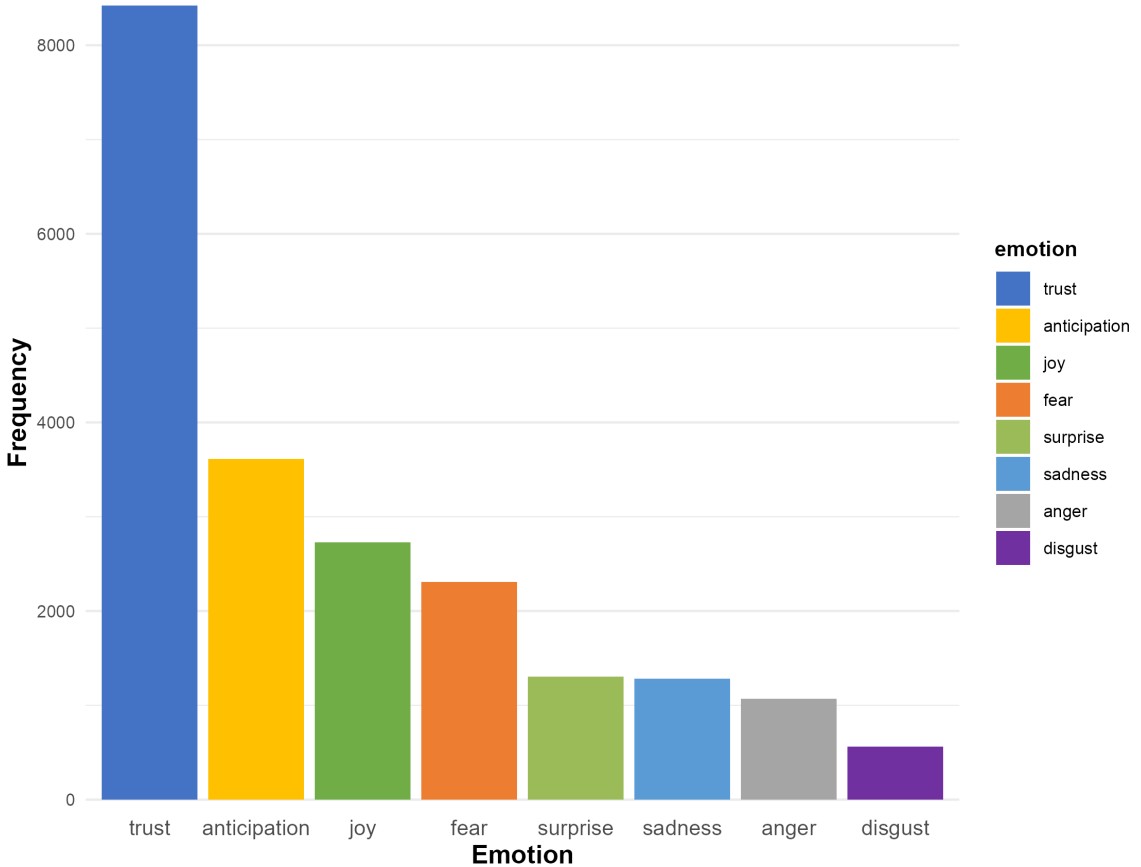

**Fig 8. Emotion distribution.**

ChatGPT. The relatively low occurrence of anger (around 1,200 instances) and disgust (around 700 instances) suggests that overtly negative sentiments are limited in academic discourse.

The overwhelming dominance of trust, along with the prevalence of anticipation and joy tend to reflect an optimistic and constructive tone in academia's evaluation of ChatGPT, possibly suggesting a widespread recognition of its potential benefits and a willingness to explore its applications. The presence of fear may signal areas of caution and concern, which could be related to ethical implications or misuses.

The findings shown in Fig 8 appear to be further supported by the overall sentiment distribution (positive, neutral, and negative) in the abstracts (Fig 9). While the negative sentiment distribution (9.78%) indicates the presence of critical evaluation and ethical considerations, the presence of neutral sentiment is likely to reflect scholarly rigor and objectivity. The sentiment distribution is generally aligned with the emotion distribution and may offer valuable insights into the overarching sentiment trends within academic discourse surrounding ChatGPT.

While Fig 9 provides an internal consistency check through the aggregation of eight basic emotions into sentiment polarities (positive, negative, and neutral), we conducted a robustness check of the sentiment results by using a custom-built, domain-sensitive sentiment lexicon tailored to scholarly discourse. This lexicon was manually constructed with reference to prior studies on evaluations in academic texts [15–17] and supplemented by high-frequency evaluative terms observed in our corpus. Terms were categorized into three groups: positive (e.g., significant, effective), negative (e.g., limitation, challenge), and neutral-but-relevant (e.g., study, analysis), to reflect common evaluative tendencies in

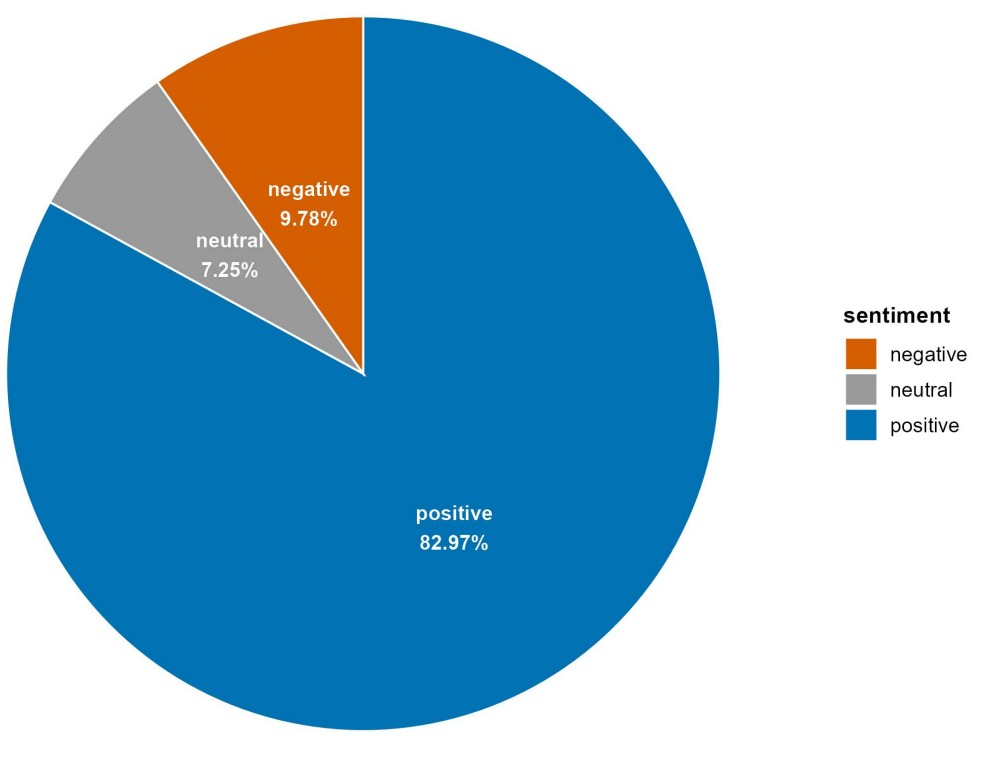

Based on analysis of 1,227 SSCI-indexed journal abstracts

**Fig 9. Sentiment distribution.**

academic writing. The scoring procedure was performed using Python 3.10 [83] in Jupyter Notebook [84]. The resulting sentiment distribution revealed an affective profile with a modest positive skew, which is broadly consistent with the sentiment polarity distribution derived from the NRC-based analysis. The results of this validation are provided in the Supporting Material (S4 Fig).

**4.3.2. Sentiment interpretation across contexts.** Although emotional expression in academic abstracts is necessarily limited by genre conventions, the emotions such as trust and anticipation could be seen as indicators of how the academic community strategically aligns itself with emerging technologies. In this context, sentiment analysis does not simply capture overt affect but reveals how academic discourse encodes shared values, expectations, and evaluative attitudes through constrained emotional cues.

To further contextualize these findings, we compare sentiment patterns in academic abstracts with those observed in other studies across educational and public domains. Our findings about sentiments are consistent with several recent studies in the educational context. For instance [85], observed that academic research articles generally maintain a slightly positive sentiment while emphasizing neutrality and objectivity. This pattern is further corroborated by [86], who found academic discussions on social media platforms showed 4.5 times more supportive than opposing views towards ChatGPT. However, sentiment distributions appear to vary significantly across different contexts and platforms. While studies by [48,86–89] consistently found positive sentiment dominance in educational settings [4], analysis of higher education faculty sentiments on Twitter presented a different pattern, with neutral sentiments (51%) surpassing positive ones (40%). This variance might be attributed to the platform's characteristics, the specific time frame of the study, or the particular concerns and perspectives of faculty members.

Interestingly, public perceptions of ChatGPT exhibit significant variations across different platforms. [51]'s analysis of Chinese social media (Weibo and Bilibili) revealed a predominantly negative sentiment (55.92%), contrasting sharply with [52]'s findings on Reddit, where 61.6% of users expressed positive sentiments. This disparity could be attributed to various factors, including varying levels of AI literacy and distinct societal concerns about AI technology across different social groups. The findings from [3] further support the generally positive public perception observed in Western social media (e.g., Twitter).

These various findings suggest that sentiment towards ChatGPT is likely to be context-dependent, varying across professional and cultural boundaries. The predominantly positive sentiment found in academic paper abstracts might reflect the academic community's recognition of ChatGPT's potential benefits while indicating a degree of critical awareness of its limitations and ethical implications. Such a seemingly balanced perspective characterized by cautious optimism may be important for fostering constructive dialogue about AI integration in academic contexts. This dialogue, in turn, may help shed light on how academic discourse discursively negotiates trust, caution, and future expectations. In what follows, we turn to a detailed reflection on how these emotions are constructed across thematic domains and what they may reveal about the academic community's attitudes towards generative AI.

### 4.4. Connections between topics and emotions

To gain deeper insight into how emotional tendencies are embedded within academic abstracts, we investigated the connections between the identified topics and emotion categories using a heatmap (Fig 10).

**4.4.1. Emotional patterns across topics.** Fig 10 illustrates varied emotional patterns across the six topics within ChatGPT related research. Trust emerges consistently as the most prevalent emotion, particularly notable in topic 3 (User Perception and Adoption) at an intensity of 100%, and it is also significantly evident in topic 1 (AI and technology communication) and topic 6 (computational processes and foundations of LLMs), both at 50%, potentially reflecting scholars' positive evaluation of technical developments. Anticipation exhibits a notable intensity of 50% in topic 6, possibly indicating scholarly interest in prospective technological advancements. Moderate anticipation levels (33.3%) are observed in topic 5 (Human-Technology Interaction), which may indicate cautious enthusiasm towards emerging collaborative frameworks. Conversely, the absence of anticipation in topic 3 and topic 4 (ethics and academic challenges) might reflect these areas' stronger focus on immediate implications rather than future possibilities. Surprise is most prominent in topic 2 (education and learning tools) at 28.6%, perhaps highlighting unexpected insights or developments in pedagogical applications. Joy is moderately observed in topic 1 (16.7%) and topic 2 (14.3%), which may reflect the academic community's favorable attitude towards innovation and practical uses.

Fear primarily emerges in topic 4 at 33.3% and somewhat less prominently in topic 5 at 16.7%, thus suggesting heightened concerns surrounding ethical risks and practical interaction challenges. Anger uniquely appears in topic 4 (33.3%), which is likely to indicate specific frustrations or critical attitudes related to ethical dilemmas surrounding the use of ChatGPT. The absence of anger across the remaining topics might suggest that while ethical issues can elicit stronger negative emotions, discussions around technical, educational, and adoption-oriented aspects tend to be approached with greater neutrality or without overt emotional charge. The complete absence of sadness and disgust across all topics may offer valuable insight into the overall emotional tenor of the academic discourse. This pattern suggests that, while challenges and limitations are acknowledged, scholars do not appear to express profound disappointment towards ChatGPT technology. Instead, this could point to a generally constructive and solution-oriented orientation in addressing the complexities of AI integration.

**4.4.2. Contextualization of emotional trends.** In our study, the predominance of trust across all topics could be meaningfully interpreted through the lens of academic discourse conventions, particularly within the abstract genre. In this context, trust represents an epistemic emotion that aligns with the academic imperative to demonstrate

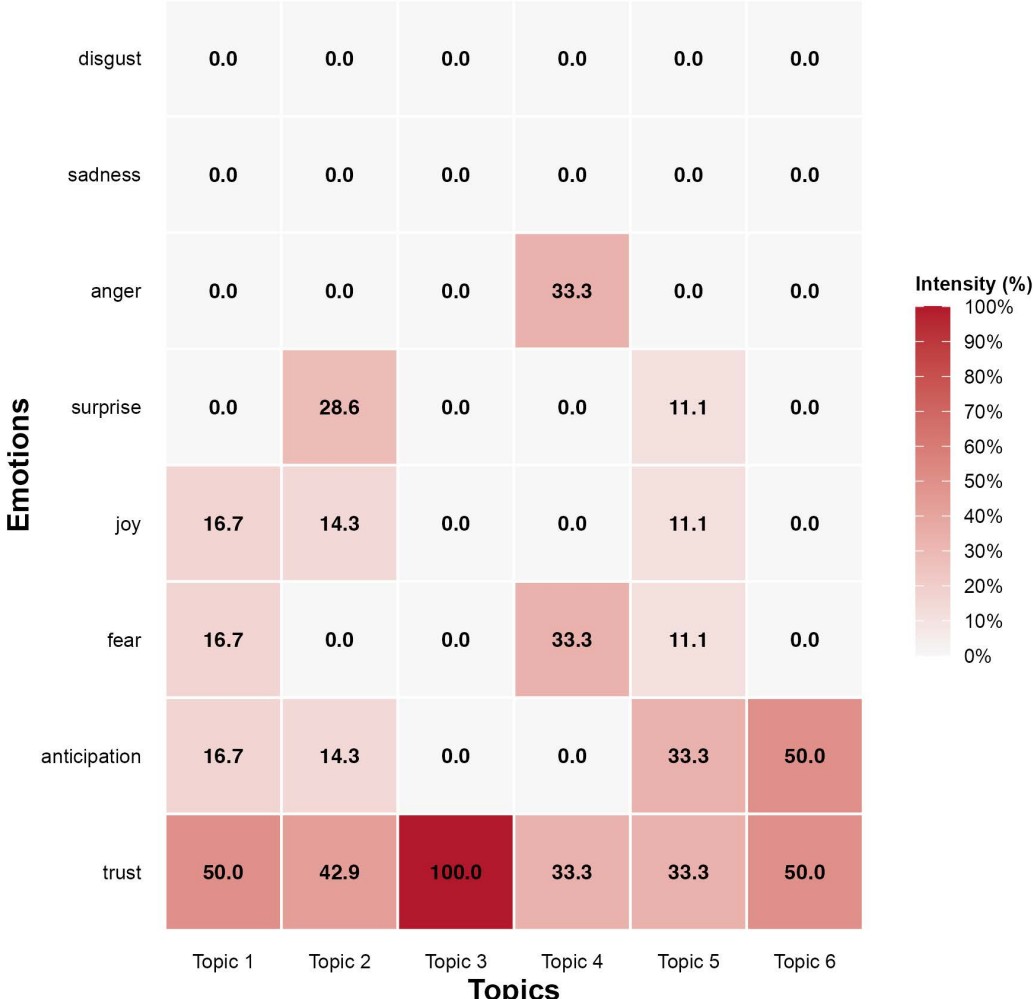

**Fig 10. Emotion distribution across topics.**

methodological rigor or conceptual legitimacy. This aligns with prior studies showing that academic writers often adopt epistemic attitudes to establish credibility and present their claims as trustworthy [15–17]. Another possible explanation of trust as the most dominant emotion across topics might be that it indicates the academia's confidence in ChatGPT's reliability and utility. This is consistent with earlier studies (e.g., [4,87,88,90]), where trust was identified as a key indicator of optimism regarding generative AI's role in enhancing creativity, engagement, and accessibility. The prevalence of trust suggests that scholars view ChatGPT as a tool that could help streamline research processes, foster innovation, and enhance pedagogical practices. Notably [4], observed that trust was especially pronounced in contexts involving higher education, where ChatGPT is perceived as a resource for improving teaching and learning outcomes.

The heightened levels of surprise and joy in topic 2 point to the academic community's recognition of ChatGPT's innovative potential in reshaping educational practices. Researchers may be discovering unexpected benefits, such as personalized learning opportunities or enhanced feedback mechanisms, which align with findings by [90] and [91]. These studies emphasize ChatGPT's role in advancing accessibility and individualization in education, thereby contributing to

its positive reception in this domain. In addition, studies like [92] and [93] both emphasize ChatGPT's ability to enhance teaching methodologies and facilitate student learning.

The high intensity of anticipation in topic 5 and topic 6 reflects the excitement about ChatGPT's evolving capabilities. According to [86], educators were particularly more emotionally expressive than general users for their opinions on ChatGPT and emotions such as trust and anticipation were quite prevalent in their study. The anticipation emotion in their findings revolves around ChatGPT's ability to assist in teaching, learning, and content creation, which is similar to the high anticipation in our study for topic 5 focusing on natural communication with technology, personalized assistance, and enhanced human-computer collaboration. The anticipation emotion in topic 6 highlights the excitement surrounding the powerful computational abilities of ChatGPT in knowledge domain. Although anticipation is moderate in topic 1, this might indicate that researchers are optimistic about the future of ChatGPT in communication, with a slightly more cautious outlook compared to its potential in human-technology interaction and broader applications.

The presence of negative emotions like fear in topic 4 reflects the academia's apprehension about risks such as academic dishonesty, ethical dilemmas, and unintended societal consequences. [12] and [51] also noted these concerns, highlighting issues related to plagiarism, job displacement, and data security. These concerns are also mirrored in findings of [5], emphasizing the importance of ethical safeguards, policy development, and user education. Researchers may be apprehensive about the ethical dilemmas and challenges that ChatGPT poses to academia. Their concerns might focus on the potential for manipulation, biased information, or the erosion of human critical thinking skills. Compared with the most obvious negative emotions in topic 4, fear is moderate in topic 1 and topic 5, which suggests that while researchers recognize AI tools' potential risks, they are generally optimistic about ChatGPT's potential in communication and human-technology interaction.

The minimal presence of emotions such as disgust and sadness indicates that while critical perspectives exist, overt negativity remains limited in academic discussions. This aligns with findings by [94], who observed constructive critiques rather than outright rejection of ChatGPT. The low level of sadness around six topics in our study might be attributed to several factors. First, academia's focus on the potential of generative AI tools likely fosters an optimistic perspective, as researchers prioritize exploring the potential benefits and applications of ChatGPT. Second, academic discourse tends to adopt a problem-solving approach when addressing challenges or risks, emphasizing solutions rather than dwelling on negative emotions. Finally, the novelty and rapid advancements in AI may generate excitement and curiosity among scholars, overshadowing feelings of sadness or negativity.

The lower emotional intensity observed in our findings on user perception and adoption (topic 3) suggests that this area is relatively neutral or less emotionally charged. Several factors may contribute to this phenomenon. First, topic 3 primarily involves an analytical focus on user behavior, adoption patterns, and factors influencing the integration of ChatGPT, which are less likely to evoke strong emotional responses compared to discussions on ethical concerns or future implications. Second, research on user perception and adoption may still be in its early stages, with limited data or established discourse, potentially leading to a narrower range of emotional expression. Finally, topic 3 may encompass a broad range of perspectives and experiences, which could result in a more balanced and neutral emotional tone. As research in this area progresses and more data become available, a wider spectrum of emotional responses may emerge.

## 5. Conclusion

Based on the topic modeling analysis of academic abstracts in the social sciences, our findings identify six distinct topics that comprehensively capture the scholarly discourse surrounding ChatGPT. These topics encompass both technical dimensions (computational processes, language model architectures) and societal implications (educational applications, user adoption patterns, ethical considerations, and human-technology interaction), indicating the multifaceted nature of academic inquiry into ChatGPT's impact across various social science domains.

Our sentiment analysis of academic abstracts reveals a potentially varied emotional landscape in the scholarly discourse, characterized by the predominance of trust and anticipation, alongside concerns represented by fear. This distribution, which is complemented by the overall sentiment pattern favoring positive affect, may suggest that while the academic community tend to maintain an optimistic and constructive outlook on ChatGPT's potential, academia also appears to acknowledge the importance of critical evaluation and ethical considerations in its implementation. Furthermore, a cross-topic sentiment analysis points to distinct emotional patterns across the six identified topics. Trust consistently emerges as the dominant emotion across all topics, while certain themes appear to evoke more specific emotions such as surprise in educational applications and fear in ethical considerations. This different distribution of emotions across topics suggests that although scholars generally express positive sentiments, their emotional responses are closely shaped by domain-specific challenges and opportunities, highlighting the complex nature of ChatGPT's integration into academic contexts.

### 5.1. Limitations

While this study provides data-driven insights into how academic discourse engages with ChatGPT, several limitations must be acknowledged. First, the sentiment analysis relies on the NRC Emotion Lexicon, which was originally developed for general and social media texts. Although we implemented a supplementary validation using a domain-sensitive academic lexicon, the NRC-based results may still be affected by genre-specific linguistic features that the NRC lexicon was not originally designed to capture.

Second, although academic discourse has been used in prior studies as valid sources for sentiment analysis, the constrained rhetorical structure of abstracts may limit the expression of affective or evaluative cues. In addition, the dataset analyzed in this study consists of a non-random sample drawn from abstracts of researchers who have chosen to publish on ChatGPT in SSCI-indexed journals. This introduces a potential selection bias, as these researchers may hold more engaged or optimistic views compared to the broader academic community. As such, the findings should be interpreted within the context of this self-selecting group, rather than generalized to all scholars or disciplines.

Third, our analysis focused on the social sciences as a whole without disaggregating results by sub-disciplines. As a result, potential disciplinary differences could not be explored. Moreover, the corpus primarily consisted of abstracts from journals in social science domains. While this provided a focused view of socially oriented scholarship, it may have reinforced topic patterns that are inherent to these disciplines. The generalizability of the findings to other research communities, especially STEM-related fields, remains limited.

Fourth, thematic interpretations such as the dominance of education-related terms may partly stem from genre conventions of abstract writing, where terms like "study," "research," and "learning" are inherently common. A more granular documentation of journal types and publication contexts may be needed to provide better transparency regarding corpus composition.

Lastly, while the exploratory, data-driven nature of this study enabled the identification of emergent themes and sentiment patterns, it may limit the extent to which findings are theoretically grounded. Future research could enhance explanatory depth by integrating discourse-analytic or affective-theoretical frameworks.

### 5.2. Future research directions

Building on these limitations, several promising directions for future research can be identified. First, expanding the corpus to include abstracts from STEM and interdisciplinary journals would allow for cross-disciplinary comparisons. Such an approach could help uncover whether different academic communities conceptualize and evaluate ChatGPT in divergent ways.

Second, a diachronic analysis examining how sentiment and thematic trends evolve over time could provide valuable insight into the shifting academic discourse on generative AI. As ChatGPT continues to evolve and its applications become more integrated into academic workflows, tracking longitudinal changes may help capture emerging concerns, adaptations, or acceptance patterns.

Third, incorporating mixed-methods approaches, such as combining computational techniques with close reading or discourse analysis of full-text articles, would enable more nuanced understanding of how attitudes towards ChatGPT are discursively constructed. This could also help account for context-specific interpretations, theoretical orientations, and discipline-specific rhetorical practices that are not always visible in abstract-level analysis.

Fourth, future research could implement statistical significance testing or resampling-based inference to assess whether differences in sentiment distributions across topics are statistically robust. While such tests were not feasible in the current study due to corpus distributional constraints and the exploratory orientation of our analysis, acknowledging this limitation is important, and their inclusion in future work would strengthen the empirical validity of observed patterns.

By addressing these areas, future studies can contribute to a more comprehensive understanding of how the academic community engages with generative AI and offers empirically grounded guidance for its integration across diverse academic domains.

## Supporting information

**S1 Dataset.  The dataset used for the analyses described in this study.**
(CSV)

**S2 File.  R Code.** R scripts used for data cleaning, analysis, and visualization.
(R)

**S3 File.  Python Code.** Python code used to generate the sentiment distribution shown in S4 Fig.
(IPYNB)

**S4 Fig.  Sentiment distribution based on the academic lexicon.** Note: All datasets and scripts have also been archived and are publicly available at Figshare: https://doi.org/10.6084/m9.figshare.29625773.
(PNG)

## Acknowledgments

The authors would like to thank **North China Electric Power University** for providing access to research databases, which enabled the acquisition of the data used in this study.

## Author contributions

**Conceptualization:** Yating Tao, Qian Shen.

**Formal analysis:** Yating Tao.

**Methodology:** Yating Tao.

**Validation:** Qian Shen.

**Visualization:** Yating Tao, Qian Shen.

**Writing – original draft:** Yating Tao.

**Writing – review & editing:** Qian Shen.

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
