## [Decision Letter · Decision Letter 0]

11 Jun 2025

Dear Dr. Shen,

Thank you for submitting your manuscript to PLOS ONE. After careful consideration, we feel that it has merit but does not fully meet PLOS ONE’s publication criteria as it currently stands. Therefore, we invite you to submit a revised version of the manuscript that addresses the points raised during the review process.

**The reviewers has completed the reviewing work about your paper, note that the number of evaluated reviewers is more than two, so please make a comprehensive revision to the paper corrreponding to all the comments they raised. And Hope you can re-submit your revised edition in the next 80 days (about 2.5 months).**

We look forward to receiving your revised manuscript.

Kind regards,

Weiqiang (Albert) Jin, Ph.D.

Academic Editor

PLOS ONE

**Journal Requirements:**

1. When submitting your revision, we need you to address these additional requirements. Please ensure that your manuscript meets PLOS ONE's style requirements, including those for file naming. The PLOS ONE style templates can be found at https://journals.plos.org/plosone/s/file?id=wjVg/PLOSOne_formatting_sample_main_body.pdf and https://journals.plos.org/plosone/s/file?id=ba62/PLOSOne_formatting_sample_title_authors_affiliations.pdf 2. Please note that PLOS ONE has specific guidelines on code sharing for submissions in which author-generated code underpins the findings in the manuscript. In these cases, we expect all author-generated code to be made available without restrictions upon publication of the work. Please review our guidelines at https://journals.plos.org/plosone/s/materials-and-software-sharing#loc-sharing-code and ensure that your code is shared in a way that follows best practice and facilitates reproducibility and reuse.

**Additional Editor Comments:**

The reviewers has completed the reviewing work about your paper, note that the number of evaluated reviewers is more than two, so please make a comprehensive revision to the paper corrreponding to all the comments they raised. And Hope you can re-submit your revised edition in the next 80 days (about 2.5 months).

Reviewers' comments:

Reviewer's Responses to Questions

**Comments to the Author**

1. Is the manuscript technically sound, and do the data support the conclusions?

Reviewer #1: Yes

Reviewer #2: Partly

Reviewer #3: Partly

Reviewer #4: Partly

Reviewer #5: Yes

2. Has the statistical analysis been performed appropriately and rigorously?

Reviewer #1: Yes

Reviewer #2: N/A

Reviewer #3: Yes

Reviewer #4: No

Reviewer #5: Yes

3. Have the authors made all data underlying the findings in their manuscript fully available?

Reviewer #1: Yes

Reviewer #2: Yes

Reviewer #3: Yes

Reviewer #4: Yes

Reviewer #5: Yes

4. Is the manuscript presented in an intelligible fashion and written in standard English?

Reviewer #1: Yes

Reviewer #2: Yes

Reviewer #3: Yes

Reviewer #4: Yes

Reviewer #5: Yes

**Reviewer #1: ** I have carefully read your paper and I have some suggestions to help improve it.

First, the topic is very timely and important. Exploring how the academic community views ChatGPT is a valuable contribution to the current academic discussion. The use of topic modeling and sentiment analysis is also a great choice for your research, and the data set is robust with 1,227 abstracts, which gives your conclusions a solid foundation.

However, I think it would be helpful to include more details about the parameters α and β in the LDA topic modeling section. These parameters can influence the results of topic extraction, so clarifying them would help other researchers better understand and replicate your methods. Additionally, while the NRC EmoLex dictionary is widely used in sentiment analysis, it wasn’t specifically designed for academic texts. It would be good to discuss its applicability and limitations in this context.

In terms of interpreting the results, I think there could be more analysis on how Topic 6 (on computational processes and LLMs) connects with other topics. This technical topic could potentially support or relate to more application-oriented themes, and exploring this connection could add depth to your analysis. Also, while "trust" as an emotion dominates across all topics, I think it would be useful to explore why this is the case. This could be tied to the objectivity required in academic writing, and explaining this would help clarify the findings.

On the methodology, you should acknowledge the limitation of analyzing only abstracts instead of full texts. While abstracts provide a quick overview, they might miss important details, which could affect the comprehensiveness of the results. There’s also no discussion about validating the topic modeling results. It would be beneficial to include an evaluation of topic consistency or even some manual validation to ensure the topics are representative and coherent.

To boarder the research scope of the related work contents in this article, if the author think it is convenient, please cite the following valuable literature (which are all with the topic of sentiment analysis).: [1] WordTransABSA: Enhancing Aspect-based Sentiment Analysis with masked language modeling for affective token prediction [2] Back to common sense: Oxford dictionary descriptive knowledge augmentation for aspect-based sentiment analysis [3] Using Masked Language Modeling to Enhance BERT-Based Aspect-Based Sentiment Analysis for Affective Token Prediction.

Regarding the literature review, the section on public attitudes toward chatbots (Section 2.3) could be organized more clearly. I suggest grouping the studies by platform type (e.g., Twitter, Reddit, Weibo) or sentiment (positive, negative, neutral) for a more systematic review. Also, there is no discussion on how different versions of ChatGPT (such as GPT-3.5 vs. GPT-4) might lead to differences in academic views. This could be an important point to address.

Lastly, for the figures, Figure 8 (the relationship between topics and sentiment) could be made clearer. I recommend considering a heatmap instead of a radar chart to better show the strength of the relationship between sentiment and topics. Adding a network graph to display the connections between the six topics would also be useful to visualize the potential links among them.

Overall, your paper is solid, and I hope these above suggestions are meant to help improve its quality and impact. I hope this feedback is helpful to you.

**Reviewer #2: ** This study investigates scholars' perceptions of ChatGPT in social sciences using topic modeling and sentiment analysis based on 1,227 SSCI abstracts. The manuscript is generally well organized and covers a timely topic. However, the study remains largely descriptive, the theoretical contribution is limited, and several methodological and presentation issues need to be addressed. Overall, this work is complete, but there is still room for improvement, my suggestions are as follows:

1. The abstract mechanically lists methods and results without explaining the novelty or importance. You should revise the abstract to emphasize what the study contributes beyond previous public/social media studies on ChatGPT.

2. The authors mention manual filtering of abstracts but do not provide inclusion/exclusion criteria (e.g., how irrelevant abstracts were defined). Please explicitly describe the manual screening criteria and report how many abstracts were excluded at each stage.

3. There is no measure of topic coherence reported to verify whether the topics make sense semantically. You should report topic coherence scores or conduct a manual validation where human coders check the interpretability of extracted topics.

4. The manuscript briefly mentions using ldatuning metrics to select the number of topics but does not show the evaluation curves or the rationale behind choosing six topics beyond metric convergence. Please provide the full tuning plots (even in supplementary materials) and justify why 6 topics were selected instead of neighboring numbers (e.g., 5 or 7).

5. NRC is designed mainly for general or social media text and may not fit the tone of academic abstracts, leading to biased sentiment results. You should discuss this limitation.

6. Figures lack statistical significance testing, emotion distributions across topics are presented visually, but no statistical tests (e.g., chi-square, ANOVA) are conducted to validate whether differences are significant.

7. Sentiment analysis lacks robustness checks, only a single pass of NRC-based analysis was performed. Please add at least one robustness check or sensitivity analysis to validate sentiment classification outcomes.

8. This manuscript claims that scholars have an “optimistic and constructive” view of ChatGPT; however, the analysis only covers abstracts, not full articles or behavior. You should tone down the strength of conclusions.

9. The following recent papers related to ChatGPT should be discussed:

[1] J Li, Y Yang, R Chen, D Zheng, PCI Pang, CK Lam, D Wong and Y Wang. 2025. Identifying Healthcare Needs with Patient Experience Reviews Using ChatGPT. PLOS One, 20(3), e0313442. doi:10.1371/journal.pone.0313442

[2] Hosseini, Mohammad, et al. "An exploratory survey about using ChatGPT in education, healthcare, and research." Plos one 18.10 (2023): e0292216.

Thank you for the opportunity to review this paper. I look forward to a better version of this manuscript.

**Reviewer #3:**  Title: scholars’ perceptions by reviewing abstracts of papers.

Getting more insight in ways that ChatGPT is addressed in social sciences research can add relevant insights to better understanding what ChatGPT can contribute in a research context. Therefore, the topic of the manuscript is interesting to address. However, the manuscript needs considerable improvements to make the main message consistent. Two main points are that the sample (the abstracts) do not necessarily reflect scholars’ perceptions since the studies may not reflect perceptions at all and an abstract is a structured text describing a research result, and not necessarily a scholarly attitude or emotion (or preferably not an emotion). In addition, a sentiment analysis may therefore not be the most appropriate method to apply to abstracts. Therefore, I recommend major revisions which can improve the manuscript. In the remainder below, I will discuss the manuscript and provide suggestions for improvements.

The ms deals with a key thematic patterns and academic perspectives in social sciences studies on ChatGPT. Via Latent Dirichlet Allocation (LDA) topic modelling and sentiment analysis techniques more than 1200 abstracts from studies in the social sciences have been analysed. This is an impressive sample, however, in my view, the study has various limitations which are not sufficiently addressed yet. Therefore, I believe that the current conclusions cannot be supported by the findings and I would advice major revisions to improve the manuscript.

More general about the text:

Generally, the manuscript is well-written with few spelling mistakes and generally good use of English (a few tenses can be improved). The structure of the manuscript is clear and for most part logical.

Not always references are added at places where these would be expected. I’ve indicated a few examples below.

Some suggestions per section

Title: The title Scholars’ Perceptions of ChatGPT in Social Sciences is somewhat misleading since the analysis was applied to the abstracts of the studies. Abstracts do strictly speaking not show scholars’ perceptions of ChatGPT, they report the outcomes of studies. I would advise to adapt the title so that it fits the study content.

Abstract: The conclusions that “the findings offer valuable insights into scholarly attitudes towards ChatGPT within the social sciences” needs more precise wording: As said abstracts do not necessarily reflect scholarly attitudes, they rather reflect outcomes of studies which should be objectively phrased (so often neutral), and therefore the conclusions cannot be supported. The abstracts provide structured summaries of the studies, however, no selection of particular groups of social studies have been made; and therefore the link between the conclusions and the method is not strong. Additionally, the abstract of the manuscript raises questions why only social studies abstracts have been included. It could have been interesting to compare abstracts from various domains (like social sciences and medical sciences or natural sciences) and compare main themes for each domain and how these differ. Also, after reading the abstracts, immediately, the question is raised why the analysis was limited to abstracts only. Abstracts, in general, have a certain function to attract the readers’ interest and therefore, negative outcomes, or more nuanced information about the precise outcomes of the studies will be lacking, amongst others due to the allowed number of words and the function and systematic structure of the abstracts. I would even argue that it is rather the case that often studies with negative outcomes will not be accepted by journals which will give the abstracts a biased representation. Thus, the usefulness of the sentiment analysis can be debated. Overall, the link between scholarly attitudes and ChatGPT needs more explanation.

Keywords: do the keywords reflect the main topics of the study? Key in the abstract and title are the perceptions or attitudes?

1. Introduction:

- Is the correct use of English applied, e.g. p1: “chatgpt has been constantly revolutionized”…is that a correct tense?

- P8: “this optimism is tempered by warnings that the risks of misuse, if unchecked, may overshadow the benefits” > can a reference be added?

- P8: “but it also introduces risks such as plagiarism and ethical breaches” > can a reference be added.

- I’m wondering whether the intro of benefits and risks is sufficiently detailed. The description is limited to a few benefits and a few risks. I would be interested here to include a more detailed and systematic discussion and an overview of the (most important) benefits and risks, and of course, supported by literature (studies). P9: the discussion of how various research fields within social sciences see the development is useful but it could benefit from a more systematic discussion. The sub fields are not clear and the discussion looks somewhat randomly. And the discussion of methods studying developments needs references.

- I appreciate the discussion of possible solutions and topics that should be addressed.

- The study addresses the gap of a systematic and data-driven analysis of studies to tract academic discussions and stance on ChatGPT. The formulation in this sentence is more in line with the methodology than how it is described in the title and abstract (scholarly attitudes or perceptions). However, in the next sentence, the aim is not in line with the gap that is identified. The aim according to the manuscript (p10) is: “examining the key debates and academia’s attitudes on ChatGPT’s role in academic settings”.

- A motivation could be added for the choice to limit the analysis to the social sciences.

- Indeed topic modelling and sentiment analysis can be used for thematic analysis and providing insights, however a better description of the situations when these methods can be applied is useful since the methods for the studies mostly are applied to media articles which contain a different type of text than abstracts. As said above, abstracts are typically structured in a similar way and do not leave much room for emotions. The question is really whether an analysis of abstract can provide the desired insights since abstracts. In the Method section (p10 and further) more motivation is provided. Consider when to present which information.

- It can be useful to provide the main research question at the end of the introduction section.

2. Theoretical frame:

- Indeed MML can uncover dominant themes and sentiments. However, an abstract does not reflect an attitude but abstracts represent academic outcomes preferably reported on in a neutral way. Attitudes or perceptions are not necessarily included, or rather should not be included. Moreover, abstracts are structured texts with similar information in a similar style in most of the text which is contrast to the sentiment analysis most often applied to unstructured texts to catch emotions.

- A concern is that CDA is described as sentiment analysis but the two methods are not mutually exchangeable. A qualitative CDA often analysis text in-depth and can include a sentiment analysis. But a sentiment analysis which only looks at the tone (positive, neutral or negative) is rather limited and I would rank it rather as a quantitative than qualitative method. The details and nuances a CDA can provide can not be provided with such a limited and automated sentiment analysis. The literature review addresses some aspects but is not consistent with regarding to the call to combine quantitative analysis with qualitative CDA, while in practice a quantitative sentiment analysis is performed. The argument why the sentiment analysis is able to reflect attitudes is not convincing..

- I have a sincere question regarding to what extent an analysis of abstracts of academic papers provides insights into academia’s attitudes on ChatGPT’s role in academic settings.

- The section ends with 3 RQs of which the first can be answered based on the abstracts, although the limitation to the social sciences limits the broadness of the outcomes. However, RQ2, regarding emotions of the scholars themselves towards ChatGPT cannot be answered by analysing the abstracts. The question raised by RQ3 is not sufficiently addressed in the literature review.

3. Data and methodology:

- Selection process is consistently described and applied as are steps for the analysis.

- Latent Dirichlet Allocation (LDA) was applied. All steps are described in detail. The study describes a very systematic approach for topic modelling.

- In the figures: in figure 3 filler words are used, but why are still filler words included in the results, while other filler words like ‘the’ have been removed? In that sense, figure 3 does not give really surprising results. Why is that the case? Can it be that the function of abstracts is to report on a study, and present outcomes? what can be learned from the outcomes? I would appreciate some reflection on that.

- I would suggest to add a figure with an overview of the research procedure.

4. Results and discussion:

- Study, research and education most frequent words. Interpretation from the authors that educational contexts may dominate the current scholarly discourse in social sciences. (p. 15) > Isn’t that because abstracts describe a research or study by definition and look forward to what can be learned from it? What is missing is more contextual information about what kind of journals / abstracts were included, more in detail than is provided currently. Since the corpus included mainly abstracts from educational sciences / social sciences, the question rises, how representative the results are?

- The analysis itself has been conducted very precisely and it shows that the authors are skilled to use this methodology; the way it is written up in detail is appreciated a lot. Still the results, however, make me wonder what is new or interesting since abstracts on these topics should include these words. The selection of the type of abstracts has narrowed the scope while it would have been interesting to compare abstracts from different fields, for example, but that hasn’t been done.

- To what extent are the results an outcome of the type of journals that were selected and what social sciences research always implies: “Our findings regarding AI and technology communication (Topic 1) and human technology interaction (Topic 5) underscore the academic community’s significant focus on the social and technical dimensions of ChatGPT” AND . > “reflects a growing recognition of the need to understand AI’s social implications beyond purely technical considerations” (that is what social sciences research implies?).

- It would have been interesting to compare abstracts from technical journals since likely these do not mention social aspects.

- One of the findings from the sentiment analysis is: “On the other hand, emotions such as fear, surprise, sadness, anger, and disgust are considerably less frequent” A plausible reason could be that these kind of emotions do normally not appear in abstracts which are academic texts which also can explain why trust has the highest frequency: that is something that you want to achieve with an abstract.

5. Conclusion:

- As already pointed out: a scholarly discourse is something very different from scholars’ attitudes.

- Limitation of social science abstracts is addressed (good). Make the limitations a separate heading. And discuss also other limitations such as the used method and the limitation to quantitative methods while a plea is held for combining quantitative and qualitative methods.

- Suggestions for further research: why is this research not conducted? A comparison of fields would be more interesting.

**Reviewer #4:**  It is timely and significant to investigate how generative artificial intelligence (GAI) is perceived within social sciences research. To date, no previous study has systematically examined these research questions using a large-scale cross-disciplinary literature data. This study may make a contribution.

However, I am unsure how the authors position this work. Based on the current manuscript, it appears that the study is not clearly framed either as an empirical study or as a systematic literature review. In the literature review section, the authors primarily summarise methodological approaches (e.g., topic modelling and sentiment analysis) without adequately reviewing prior research on GAI within the social sciences. This gives readers the impression of insufficient engagement with the existing body of work. If the author positions this work as an empirical study, there are several serious concerns. Most notably, there is a high degree of overlap between the stated research aims, data source, and methodological approach of this paper and those of existing systematic literature reviews. The exploration of how social sciences engage with GAI is, in fact, a common aim of literature reviews that employ LDA, STM, and other NLP methods. But compared with it, the current research questions appear overly narrow, and the discussion lacks theoretical depth. The findings are presented descriptively, without deeper interpretation or integration with the existing literature.

Despite this, I strongly recommend that the authors reframe this study as a systematic literature review rather than an empirical research article. There are several potential benefits to this repositioning: First, the high degree of overlap between the stated research aims, data source, and methodological approach of this paper and those of existing systematic literature reviews. Second, the use of computational topic modelling techniques follows the trend of current literature reviews. Third, the addition of sentiment analysis adds novelty and nuance, distinguishing the study from other reviews.

In line with the standard, this study falls short in several key aspects, and there are some critical issues that should be solved before publication. What is worrying most is that the manuscript lacks analytical depth and integration of seminal studies within each topic. The existing systematic literature review by LDA can also output deep insights, combined with qualitative analysis. For example, they cite current highly cited work to discuss the specific topic and research questions of current research, and list what work has most influenced them. Even some of them propose the shortcomings of current research and directions for future study. The authors do not have to fulfil all of them, but please pursue at least some of them to enhance the scholarly contribution. Here are several specific suggestions to improve the manuscript and strengthen the theoretical contribution to our knowledge community.

• The authors could strengthen qualitative interpretation of topic modelling results by incorporating discussion of exemplary works, influential keywords, or ongoing debates within each topic area.

• The authors can enhance critical reflection on current research.

• Regarding sentimental analysis among each topic, the author could compare how different disciplines (e.g., education science vs. sociology) vary in their thematic focus and emotional tone. For instance, are the keywords and attitudes in education research toward GAI more optimistic than in sociology? This type of analysis could uncover valuable disciplinary differences.

Additionally, some other major and minor problems require the authors’ attention.

1. In the Introduction section, the authors should rewrite it to foreground the research aim and clearly articulate the significance and necessity of the study at the outset. Background details—such as the emergence of ChatGPT and its societal impact—can be summarised here and expanded upon later (e.g., in the Literature Review section). Such restructuring would improve the readability and strengthen the significance of the research.

2. The research question presented in the Introduction should be more clearly situated within the broader landscape of social science research.

3. Based on model convergence, the authors selected a six-topic solution as the optimal choice for LDA. While this appears reasonable, a clearer justification for the selection criterion is needed. Additionally, the robustness of this choice could be strengthened by comparing topic coherence across alternative models, testing topic stability through multiple model runs, or assessing semantic exclusivity and interpretability.

4. The lexicon-based NRC sentiment analysis method has known limitations, particularly when applied to academic texts, which often employ a neutral tone even when expressing strong critique or praise. The authors should justify their methodological choice, acknowledge its limitations, and consider incorporating a more nuanced interpretation in the Discussion section.

I sincerely hope the authors will take these suggestions into consideration and make a contribution to the literature.

**Reviewer #5: ** This is a well written paper. The analysis is well conceived and executed (cool use of latent dirichlet allocation/machine learning). I recommend publication pending one major issue that needs addressing: data limitations. The authors analyze 1,227 research article abstracts and use these as a basis for claims on how scholars assess AI. However, this data represents a nonrandom subset of scholars, those who chose to spend considerable time and energy publishing on the topic. It makes sense that such a selection effect would go along with more positive results. On page 23 (pdf) the authors note that different data sources such as twitter posts yield different sentiments towards LLMs. This is glossing over what is a major limitation/issue that should be addressed throughout the text. Claims need to be qualified with the caveat that the authors’ data is a biased sample. This issue doesn’t defeat the project or mean it’s not without value (and cool), it merely should be noted by the authors as a limitation both up front and in the discussion. For example on page 12 (pdf) the authors state one RQ as “What are scholars’ general emotions towards ChatGPT?” But aren’t examining scholars in general or a representative sample of scholars, rather we have a nonrandom sample of scholars’ sentiments. Simple fix with a qualifier added to the RQ and the rest of the paper to match what is actual done on the empirical side: What are the general emotions of scholars who publish papers on LLMs towards ChatGPT? or something similar. Fixable issue so I recommend an R&R and publication once this and issues noted by the other reviewers are addressed.

**Do you want your identity to be public for this peer review?** For information about this choice, including consent withdrawal, please see our Privacy Policy

Reviewer #1: **Yes: ** Ningwei Wang

Reviewer #2: **Yes: ** PCI Pang

Reviewer #3: No

Reviewer #4: No

Reviewer #5: **Yes: ** Spencer Kiesel

---

## [Author Response · Author response to Decision Letter 1]

23 Jul 2025

We appreciate the thoughtful and constructive feedback provided by the reviewers and editor. A detailed point-by-point response has been submitted as a separate file titled “Response to Reviewers.docx.” We have made corresponding revisions to the manuscript and annotated all changes as requested.

---

## [Decision Letter · Decision Letter 1]

25 Sep 2025

Academic Discourse on ChatGPT in Social Sciences: A Topic Modeling and Sentiment Analysis of Research Article Abstracts

PONE-D-25-18900R1

Dear Dr. Shen,

We’re pleased to inform you that your manuscript has been judged scientifically suitable for publication and will be formally accepted for publication once it meets all outstanding technical requirements.

Lastly, the Reviewer #2 still has some concerns, please revise these points beforce proof the publication version of manuscript.

Kind regards,

Weiqiang (Albert) Jin, Ph.D.

Academic Editor

PLOS ONE

Additional Editor Comments (optional):

Reviewer #2:

Reviewer #4:

Reviewer #5:

Reviewers' comments:

Reviewer's Responses to Questions

**Comments to the Author**

Reviewer #2: All comments have been addressed

Reviewer #4: All comments have been addressed

Reviewer #5: All comments have been addressed

2. Is the manuscript technically sound, and do the data support the conclusions?

Reviewer #2: Yes

Reviewer #4: Partly

Reviewer #5: Yes

3. Has the statistical analysis been performed appropriately and rigorously?

Reviewer #2: N/A

Reviewer #4: Yes

Reviewer #5: Yes

4. Have the authors made all data underlying the findings in their manuscript fully available?

Reviewer #2: Yes

Reviewer #4: Yes

Reviewer #5: Yes

5. Is the manuscript presented in an intelligible fashion and written in standard English?

Reviewer #2: Yes

Reviewer #4: Yes

Reviewer #5: Yes

Reviewer #2: I have reviewed the revised manuscript and the authors point-by-point responses. Overall, the revisions address my previous concerns in a substantive manner.

Comment 1: Innovation and distinction from social-media studies: Fully addressed.

Comment 2: Screening criteria and removal counts: Fully addressed.

Comment 3: Topic interpretability/validation: Fully addressed; a small manual-coding check could be added in future.

Comment 4: ldatuning metrics and choice of six topics: Fully addressed.

Comment 5: NRC lexicon applicability and limitations: Fully addressed.

Comment 6: Statistical significance for cross-topic sentiment differences: Partially addressed; moved to “future work” but no test performed.

Comment 7: Robustness/sensitivity check: Fully addressed (added domain-sensitive lexicon).

Comment 8: Introduction focus and scope: Fully addressed.

Comment 8: Conclusion tone: Fully addressed.

Comment 9: Abstracts as source of attitudes: Fully addressed.

Comment 10: CDA and sentiment analysis clarification: Fully addressed.

Comment 11: Limits of inferring “academic attitudes”: Fully addressed.

Comment 12: Research question alignment: Fully addressed.

Comment 13: Figure 3 clarity: Fully addressed.

Reviewer #4: The authors have largely addressed the earlier reviewer feedback. I recommend acceptance pending the following minor editorial corrections:

Voice error: "All abstracts were first converted into lowercase to ensure uniformity, and remove all punctuation, numbers, and special characters were removed."

Spelling error: "LDA enables dimensional reduction … with an estimation of a sparse Drichlet prior."

Reviewer #5: I am satisfied with the authors revisions.

**Do you want your identity to be public for this peer review?** For information about this choice, including consent withdrawal, please see our Privacy Policy

Reviewer #2: **Yes: ** Patrick Cheong-Iao Pang

Reviewer #4: No

Reviewer #5: No

---

## [Editor Report · Acceptance letter]

PONE-D-25-18900R1

PLOS ONE

Dear Dr. Shen,

I'm pleased to inform you that your manuscript has been deemed suitable for publication in PLOS ONE. Congratulations! Your manuscript is now being handed over to our production team.

Kind regards,

on behalf of

Dr. Weiqiang (Albert) Jin

Academic Editor

PLOS ONE